# Using high-resolution regional climate models to estimate return levels of daily extreme precipitation over Bavaria

Benjamin Poschlod[1,a]

[1]Department of Geography, Ludwig-Maximilians-Universität München, Munich, 80333, Germany
[a]now at: Research Department, Berchtesgaden National Park, Berchtesgaden, 83471, Germany

*Correspondence to*: Benjamin Poschlod (Benjamin.Poschlod@npv-bgd.bayern.de)

**Abstract.** Extreme daily rainfall is an important trigger for floods in Bavaria. The dimensioning of water management structures as well as building codes are based on observational rainfall return levels. In this study, three high-resolution regional
climate models (RCMs) are employed to produce 10-year and 100-year daily rainfall return levels and their performance is evaluated by comparison to observational return levels. The study area is governed by different types of precipitation (stratiform, orographic, convectional) and a complex terrain, with convective precipitation also contributing to daily rainfall levels. The Canadian Regional Climate Model version 5 (CRCM5) at 12 km spatial resolution and the Weather and Forecasting Research model (WRF) at 5 km resolution both driven by ERA-Interim reanalysis data use parametrization schemes to simulate
convection. The WRF at 1.5 km resolution driven by ERA5 reanalysis data explicitly resolves convectional processes. Applying the Generalized Extreme Value (GEV) distribution, the CRCM5 setup can reproduce the observational 10-year return levels with an areal average bias of +6.6 % and a spatial Spearman rank correlation of $\rho = 0.72$. The higher-resolution 5 km WRF setup is found to improve the performance in terms of bias (+4.7 %) and spatial correlation ($\rho = 0.82$). However, the finer topographic details of the WRF-ERA5 return levels cannot be evaluated with the observation data because their spatial
resolution is too low. Hence, this comparison shows no further improvement of the spatial correlation ($\rho = 0.82$) but a small improvement of the bias (2.7 %) compared to the 5 km resolution setup.

Uncertainties due to extreme value theory are explored by employing three further approaches. Applied on the WRF-ERA5 data, the GEV distribution with fixed shape parameter (bias = +2.5 %, $\rho = 0.79$) and the Generalized Pareto (GP: bias = +2.9 %, $\rho = 0.81$) show almost equivalent results for the 10-year return period, whereas the Metastatistical Extreme Value (MEV)
distribution leads to a slight underestimation (bias = -7.8 %, $\rho = 0.84$). For the 100-year return level, however, the MEV distribution (bias = +2.7 %, $\rho = 0.73$) outperforms the GEV distribution (bias = +13.3 %, $\rho = 0.66$), the GEV distribution with fixed shape parameter (bias = +12.9 %, $\rho = 0.70$), and the GP distribution (bias = +11.9 %, $\rho = 0.63$). Hence, for applications, where the return period is extrapolated, the MEV framework is recommended.

From these results, it follows that high-resolution regional climate models are suitable for generating spatially homogeneous
rainfall return level products. In regions with a sparse rain gauge density or low spatial representativeness of the stations due to complex topography, RCMs can support the observational data. Further, RCMs driven by global climate models with

emission scenarios can project climate change-induced alterations in rainfall return levels at regional to local scales. This would allow adjustment of structural design and, therefore, adaption to future precipitation conditions.

## 1 Introduction

Extreme rainfall is an important driver for different kinds of hydrometeorological hazards, such as flooding and mass movements. The state of Bavaria is exposed to the highest daily rainfall intensities in Germany. Due to the complex topography and a dense river network the area is prone to riverine flooding and landslides (Grieser et al., 2006; Wiedenmann et al., 2016). Furthermore, urban areas are at risk of urban flooding due to the dense population and a large fraction of impervious areas (Chen and Leandro, 2019). To assess the risk of heavy precipitation events and to dimension adaptation measures, engineers

and public authorities often use the concept of rainfall return levels. In Germany, a rainfall return level database ("Coordinated heavy precipitation regionalization evaluation"; KOSTRA; Junghänel et al., 2017; Malitz and Ertel, 2015) is supplied by the German weather service, which is based on rain gauge observations. A similar product is available for Austria (Kainz et al., 2007). MeteoSwiss provides mapped return levels and pointwise data (MeteoSwiss, 2021). These products are included in building standards and are, therefore, widely used. Even though the coverage of rain gauges in Germany, Austria, and

Switzerland is relatively high, there are uncertainties due to the spatial representativeness of the measuring stations to generate an area-wide rainfall return level product. This problem applies even more on a continental scale as the rain gauge density is distributed heterogeneously over different European countries, where the available time series might be too short to capture a sufficient number of extreme events (Lewis et al., 2019).

Instead of using point-wise measurements, areal precipitation products (e.g. radar, satellite, or reanalysis products) could be

used as the basis for return level calculations. However, each of these areal precipitation products shows different limitations, which lead to uncertain or unrealistic return level estimations. Radar data (RADOLAN for Germany; Kreklow et al., 2020) and satellite products (e.g. CMORPH; Joyce et al., 2004 or PERSIANN; Hong et al., 2004) would provide the necessary temporal and spatial resolutions to capture extreme rainfall events. Yet, the temporal coverage of these products extends only to the early 2000s, which is why the sampling of extreme rainfall events is not sufficient for extreme value analysis.

Furthermore, radar estimates (Goudenhoofdt and Delobbe, 2016; Kreklow et al., 2020) as well as satellite products (Stampoulis and Anagnostou, 2012) reveal biases compared to rain gauges. Reanalysis data (e.g. E-OBS; Haylock et al., 2008; ERA-Interim; Dee et al., 2011; ERA5; Hersbach et al., 2020) would have the necessary temporal coverage, but they show systematic underestimation of the intensity of extreme precipitation events (Hu and Franzke, 2020; own calculations, not shown). Ehmele and Kunz (2019) apply a semi-physical two-dimensional stochastic precipitation model to calculate spatial homogeneous

return levels over Baden-Württemberg (Germany). However, the model needs to be calibrated with observational data and therefore relies on the high rain gauge density in the area.

Since the frequency and intensity of heavy precipitation will change due to climate change (Myhre et al., 2019; Poschlod and Ludwig, 2021; Westra et al., 2014), the use of climate models would provide the advantage of being able to estimate climate

change-induced alterations in rainfall return levels on a physical basis. However, this application requires careful validation of climate model results for historical conditions.

Regional climate models (RCMs) at 12 km spatial resolution have proven to deliver appropriate rainfall return level estimations for 3-hourly to daily duration (Berg et al., 2019; Poschlod et al., 2021; Ulbrich and Nissen, 2017). Although the results show a high spatial correlation to observational products and a low bias averaged over the area, local deviations are evident, especially in regions with complex topography (Poschlod et al., 2021). Also, the intensity of short-duration hourly rainfall extremes could not be reproduced at 12 km spatial resolution.

When communicating the results of climate model projections to local or regional stakeholders, insurance companies, and governmental authorities in the field of flood prevention, hydrological modelling, dimensioning of reservoirs, buildings, and water infrastructure, these aforementioned local biases may prevent the results from being accepted and implemented (Benjamin and Budescu, 2018). When presenting the study results (Poschlod et al., 2021; Poschlod and Ludwig, 2021) to a selection of representatives of the Bavarian Environmental Agency, local deviations in the climate model data stood in the way of further use or even implementation of the study results for adaptation measures to intensifying extreme precipitation events. Such discussion meetings at the interface between climate science and local experts with practical relevance provide valuable insight for practitioner demands. Therefore, one of the objectives of this study is to investigate whether higher-resolution climate models can reduce local biases in extreme precipitation. This could lead to a higher acceptance of extreme precipitation data based on climate models by government institutions, which would also support the implementation of adaptation measures.

For shorter rainfall durations, many studies have shown that higher-resolution RCMs, so-called convection-permitting models (CPMs), improve the reproduction of high-intensity short-duration convectional precipitation events (Brisson et al., 2016; Coppola et al., 2018; Fosser et al., 2014; Kendon et al., 2014). A spatial resolution of a few kilometres is considered necessary by the RCM community to explicitly resolve convection (Langhans et al., 2012, Panosetti et al., 2020; Prein et al., 2015), whereas at broader-resolutions parametrization schemes are applied to represent convection. However, also long-duration rainfall return levels can be influenced by convectional precipitation. In Germany, convectional rainfall contributes to the 24-hourly return level for roughly 50 % of the area (Malitz and Ertel, 2015). Therefore, CPMs are expected to improve the estimations of these return levels as well. Additionally, the higher spatial resolution enhances the representation of complex terrain (Karki et al., 2017; Langhans et al., 2012; Poschlod et al., 2018).

Hence, in this study, three different high-resolution RCMs featuring 12 km, 5 km, and 1.5 km spatial resolution and driven by 30-year reanalysis data are applied to reproduce daily 10-year and 100-year rainfall return levels over the complex terrain of the northern Pre-Alps and Alps. Based on interviews with stakeholders from the infrastructure sector and on legislative guidelines, Nissen and Ulbrich (2017) identified the 10-year return level as relevant threshold. Following this recommendation, the 10-year return level is chosen in this study as well representing "moderate extremes". However, since the insurance industry (Ehmele and Kunz, 2019) and flood protection (Schmitt and Scheid, 2020) are interested in longer return periods, 100-year return levels are calculated despite the higher extreme value statistical uncertainties. The daily duration is relevant for the

generation of riverine floods in the study area (Berghuijs et al., 2019; Keller et al., 2017; Merz and Blöschl, 2003), such as the two extreme flooding events in May 1999 and August 2005 in southern Bavaria, Austria, and Switzerland (BLFW, 2003; Grieser et al., 2006; LfU, 2007; Stucki et al., 2020) induced by high daily precipitation sums. However, the antecedent wetness state of the catchment also plays a major role in the transition of heavy precipitation to floods (Schröter et al., 2015).

The daily 10-year and 100-year return levels based on the three RCM setups are evaluated by means of an observational return level product using national datasets from Germany, Austria, and Switzerland. In a second step, different extreme value distributions and sampling strategies are applied to all climate model datasets to explore uncertainties due to extreme value theory and to investigate possible improvements.

The study tries to answer two main research questions: (1) Can existing RCM setups at higher spatial resolution reduce local biases and improve spatial correlation between the climate model products and the observational product? (2) How large are the differences due to the application of different state-of-the-art extreme value statistical approaches, and which approach is recommended?

## 2 Data and study area

### 2.1 Observational rainfall return level data

To evaluate the RCMs, an observation-based product is generated from the three national datasets described below. As these datasets extend to the national borders and a little beyond, the arithmetic mean is calculated in the overlapping areas. To compare gridded precipitation from the RCMs and point measurements from the observations, Breinl et al. (2020) suggest an areal reduction of 5 % for pointwise 24-hourly 10-year return levels in Austria. However, to be consistent over the study area, no areal reduction factor is applied for the daily duration following Berg et al. (2019) and Poschlod et al. (2021).

### 2.1.1 Germany

The German weather service offers gridded return level data derived from daily rain gauge measurements (Malitz and Ertel, 2015). The daily measurement window spans from 05:50 to 05:50 UTC. The observations cover a period of maximum 1951 – 2010, where only May – September are analysed as the highest rainfall amounts occur during these months. A peak over threshold (POT) sampling strategy was applied for 2231 rain gauges, where the threshold corresponds to the available time period. A maximum of 2.718 events per year on average was considered. For these samples, an exponential distribution was fitted. The resulting daily return levels are increased by 14 % to provide 24-hourly moving window estimates (Malitz and Ertel, 2015). The rainfall return levels were spatially interpolated over Germany at roughly 8 x 8 km² resolution. An uncertainty range of 15 % (20 %) is assumed for the 10-year (100-year) return levels, which is induced by measurement errors, uncertainties of the extreme value statistics and regionalization, and the internal variability of the climate system (Junghänel et al., 2017). Data are accessed from DWD (2020). As the daily return levels were beforehand transferred to 24-hourly moving window estimates, I reduce these values by 14 % to obtain daily estimates. This relation between daily fixed windows and 24-

hourly moving windows has also been applied by Poschlod et al. (2021) following Barbero et al. (2019) and Boughton and Jakob (2008).

### 2.1.2 Austria

The Austrian dataset follows a similar approach as the German dataset also applying POT sampling at 141 ombrographs (5-min resolution) and 843 ombrometers (daily resolution: 06:00 to 06:00 UTC) spatially interpolated to gridded return levels at 6 x 6 km² resolution (BMLRT, 2018). As the rain gauges are distributed inhomogeneously yielding too low return level estimations, the "orographic convective model" OKM (Lorenz and Skoda, 2001) was employed to support the observations (Kainz et al., 2007). The resulting design rainfall is based on a combination of the observational data and the weather model simulations. Further details can be found in Kainz et al. (2007) and BMLRT (2006; 2018). Data are accessed from BMLRT (2020). Again, this data product provides moving window 24-hourly estimates, which is why the 24-hourly return levels are adjusted to daily values applying a reduction of 14 % (see Sect. 2.1.1).

### 2.1.3 Switzerland

 MeteoSwiss (2021) provides pointwise daily rainfall return levels at 336 rain gauges. The daily measurement extends from 05:40 to 05:40 UTC. The observations cover the time period from 1966 to 2015. To increase the sample size, seasonal maxima were extracted and assumed to follow a Generalized Extreme Value (GEV) distribution. The GEV distribution is fitted via Bayesian estimation and the according return levels are generated (Fukutome et al., 2015). Since an areal comparison product is to be produced in this study, these point return levels are regionalised by means of ordinary kriging.

### 2.2 Climate model data

Three different RCM setups are used. The Canadian Regional Climate Model version 5 (CRCM5) driven by ERA-Interim, the Weather and Research Forecasting Model (WRF; Skamarock et al., 2008) driven by ERA-Interim, and the WRF driven by ERA-5. The selection of these three different setups was based on the following considerations: The CRCM5 driven by a global climate model ensemble has proven to reproduce rainfall return levels over Europe with good skill (Poschlod et al., 2021). However, the study has shown that internal climate variability has major impacts on the estimation of return levels. Using reanalysis data as boundary conditions strongly reduces this source of uncertainty when comparing with observation-based return levels. As described in Section 1, the resulting return levels of this RCM driven by a global climate model ensemble were presented to local authorities, but local biases prevented further implementation of the results. Therefore, the CRCM5 setup serves as a benchmark. The WRF ERA-INTERIM at 5 km resolution represents a setup optimised for the study area with higher spatial resolution but parameterisation of convection. The WRF ERA5 is the highest resolution setup available with 1.5 km resolution and calculates convection explicitly. All three climate model rainfall data sets are openly available.

### 2.2.1 CRCM5 ERA-INTERIM

The CRCM5 at 0.11° resolution equalling roughly 12 km is driven by ERA-Interim reanalysis data (Leduc et al., 2019). No nesting was applied, as with the RCM setups presented in Kotlarski et al. (2014), which are also driven by ERA-Interim and have a spatial resolution of 0.11°. Convectional processes are parametrized due to the spatial resolution. Processes related to deep convection are calculated with the parametrization scheme by Kain and Fritsch (1990). The Kuo transient scheme (Bélair et al., 2005; Kuo, 1965) is applied to represent shallow convection. A more detailed documentation of the model setup and options used is given by Hernández-Díaz et al. (2012) and Martynov et al. (2012). Daily rainfall sums of 30-year time period of 1980 – 2009 are extracted for this study.

### 2.2.2 WRF ERA-INTERIM

The WRF version 3.6.1 is set up in nested domains of 45 x 45 km², 15 x 15 km² and 5 x 5 km² spatial resolution in its non-hydrostatic mode and driven by ERA-Interim reanalysis data at 75 x 75 km² spatial resolution and 6-hourly temporal resolution (Warscher et al., 2019). Spectral nudging is applied to reduce deviations from the large-scale forcing patterns in the reanalysis data (Wagner et al., 2018). Convection is parametrized with the Grell-Freitas scheme (Grell and Freitas, 2014). The detailed model setup as well as an evaluation of different climate variables is given in Warscher et al. (2019). Here, daily rainfall data of the highest-resolution domain are used for the time period of 1980 – 2009. Data are accessed from Warscher (2019).

### 2.2.3 WRF ERA5

The WRF model version 4.1 is configured with two one-way nested domains of 7.5 x 7.5 km² and 1.5 x 1.5 km² grid spacing centred over Bavaria (Collier and Mölg, 2020). The model is forced at the outer lateral boundaries by ERA5 reanalysis data at 30 x 30 km² spatial resolution and 3-hourly temporal resolution applying spectral nudging. The higher-resolution 1.5 km setup is assumed to explicitly resolve convection, and therefore no parametrization scheme is applied. The 30-year simulation was divided into 30 annual slices starting at 1 September of each year. As the model is forced by the lateral boundary conditions at 3-hourly resolution, slicing the simulation period is not assumed to have a systematic impact on the magnitude of rainfall return levels. A detailed description of the model setup and evaluation of various climate variables is provided in Collier and Mölg (2020). However, the authors emphasize that the applied schemes and the model configuration has not been optimized for the study area due to the high computational expenses of the high-resolution run. The physics and dynamics options used in the simulations are based on former convection-permitting WRF applications (e.g. Collier et al., 2019). In this study, daily rainfall sums from 1988 – 2017 are extracted from the climate model data accessed from Collier (2020). The 1.5 km domain covers $351 \times 351$ grid cells, whereby the outer 40 cells are discarded on all sides to exclude boundary effects (Collier and Mölg, 2020).

**2.3 Description of the study area**

The area of investigation is given by the analysis domain of the highest-resolution RCM, which is centred over the state of
Bavaria, and the available observational rainfall return level data (see Fig. 1). It covers south-eastern Germany, north-western Austria, north-eastern Switzerland and Liechtenstein. The area shows altitude levels below 100 m in the northwest in the Rhine plain up to altitudes above 2500 m in the Alps. It covers various low mountain ranges, including the Ore Mountains, Odenwald, Swabian Jura and Bavarian Forest. The patterns of annual mean precipitation are governed by the complex topography (see Fig. 2; Haylock et al., 2008). Different rainfall types (convectional, orographic, stratiform) contribute to this precipitation
climatology (Malitz and Ertel, 2015). The lowest annual precipitation sums amount to 500 – 700 mm in the north of the study area. The low mountain ranges induce orographic lifting leading to precipitation sums of 1000 to 1500 mm per year. The highest precipitation sums of more than 2000 mm are found in the Alps, with dry valleys, such as the Inn valley having totals below 1000 mm. Annual average temperatures range from less than 0°C in the Alps to 10°C in northern Bavaria (DWD 2021, ZAMG 2021).


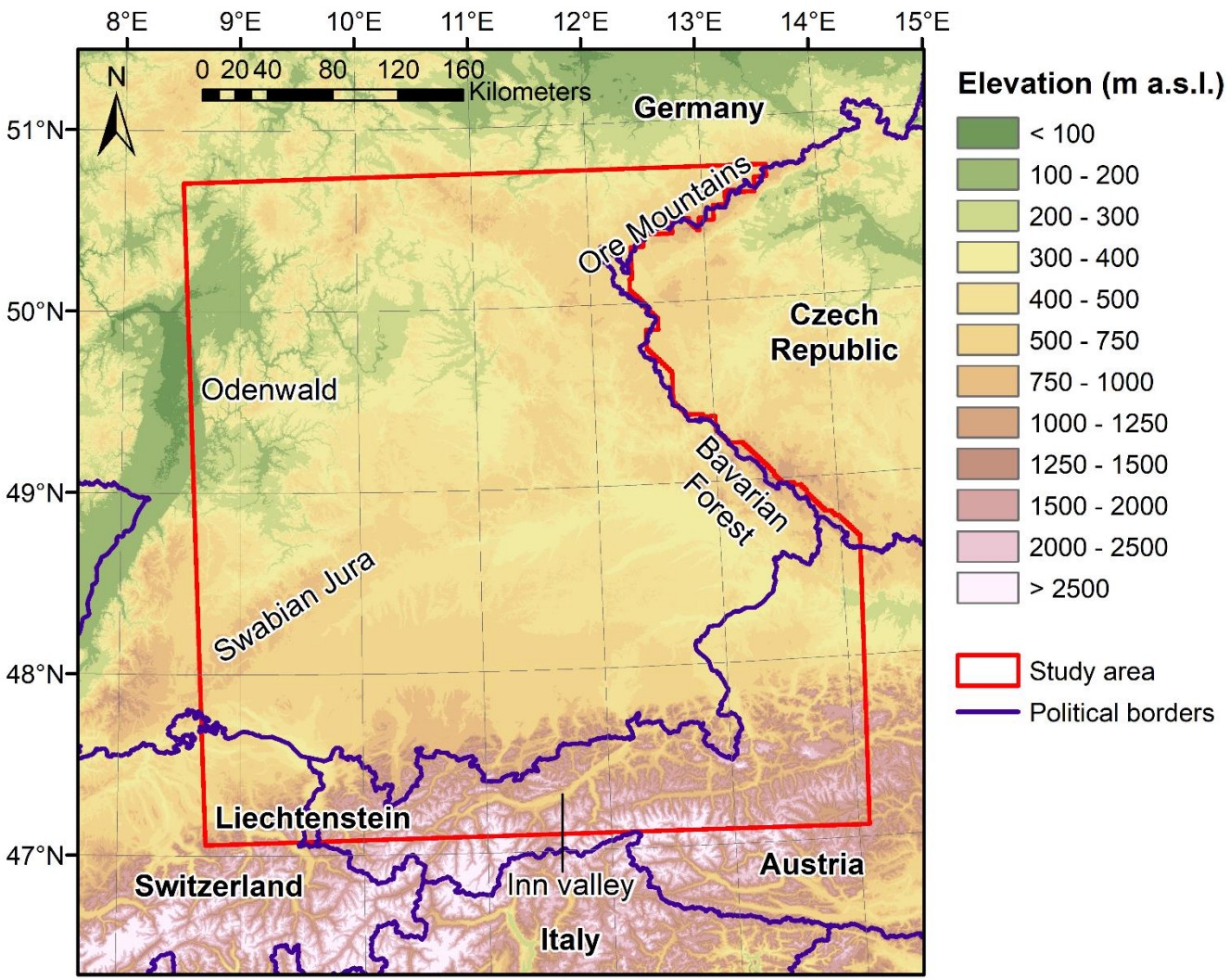

Figure 1: Topography of the investigated area. The elevation is based on the SRTM at 90 m resolution (Jarvis et al., 2008).

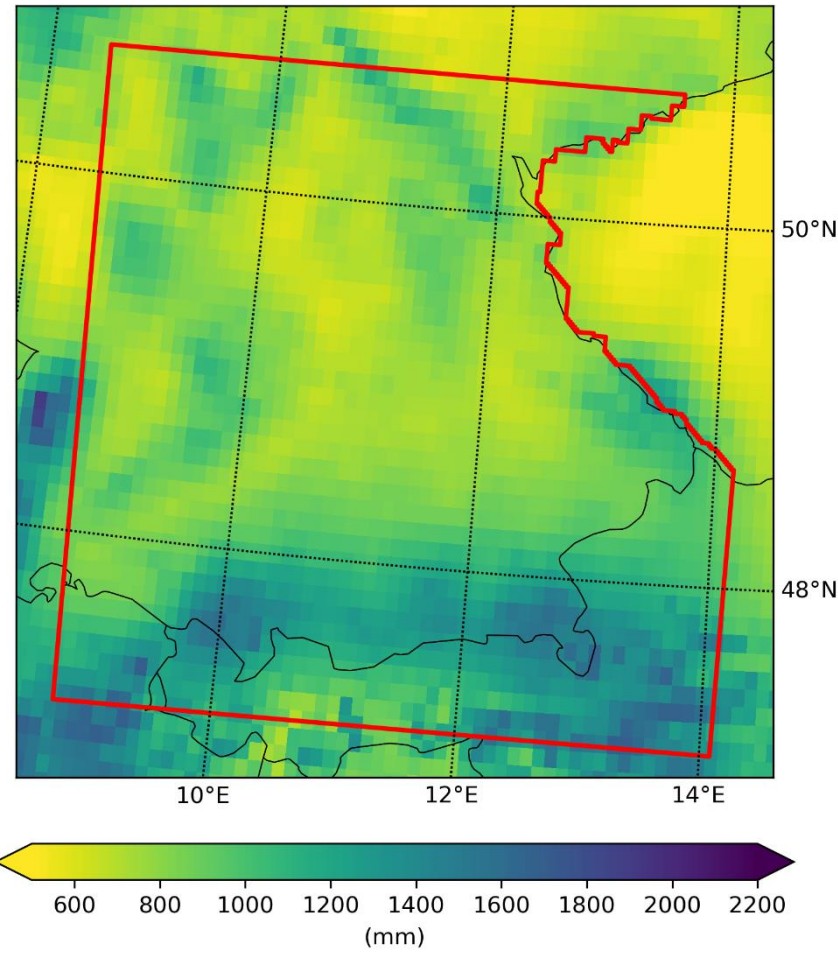

**Figure 2: Annual mean precipitation for the period 1980 – 2009 based on E-OBS (Haylock et al., 2008).**


# 3 Extreme value approaches

Extreme value theory (EVT) is applied to quantify the stochastic behaviour of a process at unusually large or small values. It is commonly used to calculate return levels for different rainfall durations (Coles, 2001).

## 3.1 Block maxima

A classical approach to sample unusual ("extreme") rainfall intensities is given by the block maxima (BM) approach (Coles, 2001). Therefore, a single value is extracted from a typically seasonal or annual block. This strategy ensures that the samples

are distant from each other leading to very low serial dependence. However, not all sampled values might be extreme. Also, the information of more than one extreme value per block is lost as these values are discarded.

The Fisher-Tippett-Gnedenko theorem (Fisher and Tippett, 1928; Gnedenko, 1943) states that the distribution of the block maxima samples tends to follow the GEV distribution, where the cumulative density function (CDF) $G$ is given with the sample size $n \rightarrow \infty$:

$$G(x; \xi) = \begin{cases} \exp\left(-\left[1 + \xi\left(\frac{x-\mu}{\sigma}\right)\right]^{-1/\xi}\right), \xi \neq 0 \\ \exp\left(-\exp\left(-\frac{x-\mu}{\sigma}\right)\right), \xi = 0 \end{cases} \tag{1}$$

The location parameter $\mu$ governs the centre, and the scale parameter $\sigma$ governs the spread of the GEV distribution. The tail behaviour of $G$ is defined by the shape parameter $\xi$ determining whether the GEV follows the Weibull ($\xi < 0$), Gumbel ($\xi = 0$), or Fréchet ($\xi > 0$) distribution (Gilleland et al., 2017). Hence, the GEV is a very flexible distribution. The drawback of this flexibility shows up in a high estimation variance of $\xi$ resulting in an unstable quantile estimate (Bücher et al., 2020).

For all three RCM setups, annual maxima of daily precipitation are extracted. Then for all grid cells trends were detected
applying the Mann-Kendall test at the significance level of $\alpha = 0.05$. The significance level describes the probability rejecting the null hypothesis $H_0$, given that $H_0$ is true. As the statistical test is carried out at $n$ grid cells, $H_0$ would be erroneously rejected at $n \cdot \alpha$ grid cells on average by design of the test setup (Ventura et al., 2004). The rate of these errors is referred to as false discovery rate (FDR; Benjamini and Hochberg, 1995). To control the FDR, the critical $p$-value is adjusted for multiple testing using the approach from Benjamini and Hochberg (1995) following Wilks (2016). $H_0$ is rejected at each grid cell $g$ if the $p$-
value of the test $p_g \leq p_{FDR}$, where

$$p_{FDR} = \max_{g=1,\ldots,n} \left\{ g : p_{(g)} \leq \alpha_{FDR} \cdot \left(\frac{g}{n}\right) \right\} \tag{2}$$

$p_{(g)}$ with $g = 1,\ldots,n$ are the sorted $p$-values of the statistical test for all grid cells $g$ of the study area. For $\alpha_{FDR}$ the value of $2 \cdot \alpha$ is recommended (Wilks, 2016).

No significant trends are found for the 30 sampled values at each grid cell of every RCM setup. The parameters of the GEV
distribution $G$ are optimized to the BM samples by estimating the L-moments (Hosking et al., 1985). This is carried out applying the R package "extRemes" by Gilleland and Katz (2016). Delicado and Goria (2008) recommend the method of L-moments for sample sizes of $n \leq 50$ as it is robust to outliers in the data. The Anderson-Darling test at the significance level of $\alpha = 0.05$ is applied to ensure the goodness of fit of the estimated GEV distribution at each grid cell (see Fig. S7). Again, the critical $p$-value is adjusted for multiple testing. Less than 0.15 % of all fits for all three climate model setups are rejected.
Based on these fits, the 10-year and 100-year return levels are derived. The spatial distributions of the GEV parameters are mapped in Figure 3.

There, the location parameter is governed by the topography (see Figs. 1 and 3a, d, g), where the spatial distribution of these parameters is similar for all three RCM setups. The spatial distribution of the scale parameter also corresponds to the

topography but shows more noise. The spatial distribution of the WRF-ERA-I and WRF-ERA5 are similar and show the

highest values of the scale parameter at the northern slopes of the Alps. The orography of the low mountain ranges of the Swabian Jura, Odenwald, Ore Mountains and Bavarian Forest also impacts the spatial pattern of the scale parameter (Figs. 3e and 3h). Lower values are found at the leesides of the low mountain ranges and the inner-alpine dry valleys. The spatial distribution of the scale parameter based on the CRCM5-ERA-I follows the topography less closely and shows an even noisier pattern (Fig. 3b). Some topographical features can nevertheless be recognised, such as the Odenwald and higher values in the

Pre-Alps and northern slopes of the Alps. The fitted shape parameter reveals a chaotic pattern with small patches of positive and negative values differing for the three RCM setups. This chaotic pattern corresponds to the high estimation variance of the shape parameter based on the limited available sample size of 30 annual maxima.

The histograms of the parameters are given in the Supplementary Materials (Fig. S1). An exemplary fit for the grid cell of Munich is shown in Figure S2 for all three RCM setups. This EVT approach is referred to as GEV-LMOM.

As small samples lead to high uncertainties estimating the shape parameter of the GEV distribution, Papalexiou and Koutsoyiannis (2013) recommend using a fixed value of $\xi = 0.114$. This approach is referred to as GEV-FIX. The Anderson-Darling test at the significance level of $\alpha = 0.05$ is carried out in the same way as for GEV-LMOM (see Fig. S7). Less than 0.01 % of all fits are rejected. Figure S3 provides an exemplary fit for the grid cell of Munich.

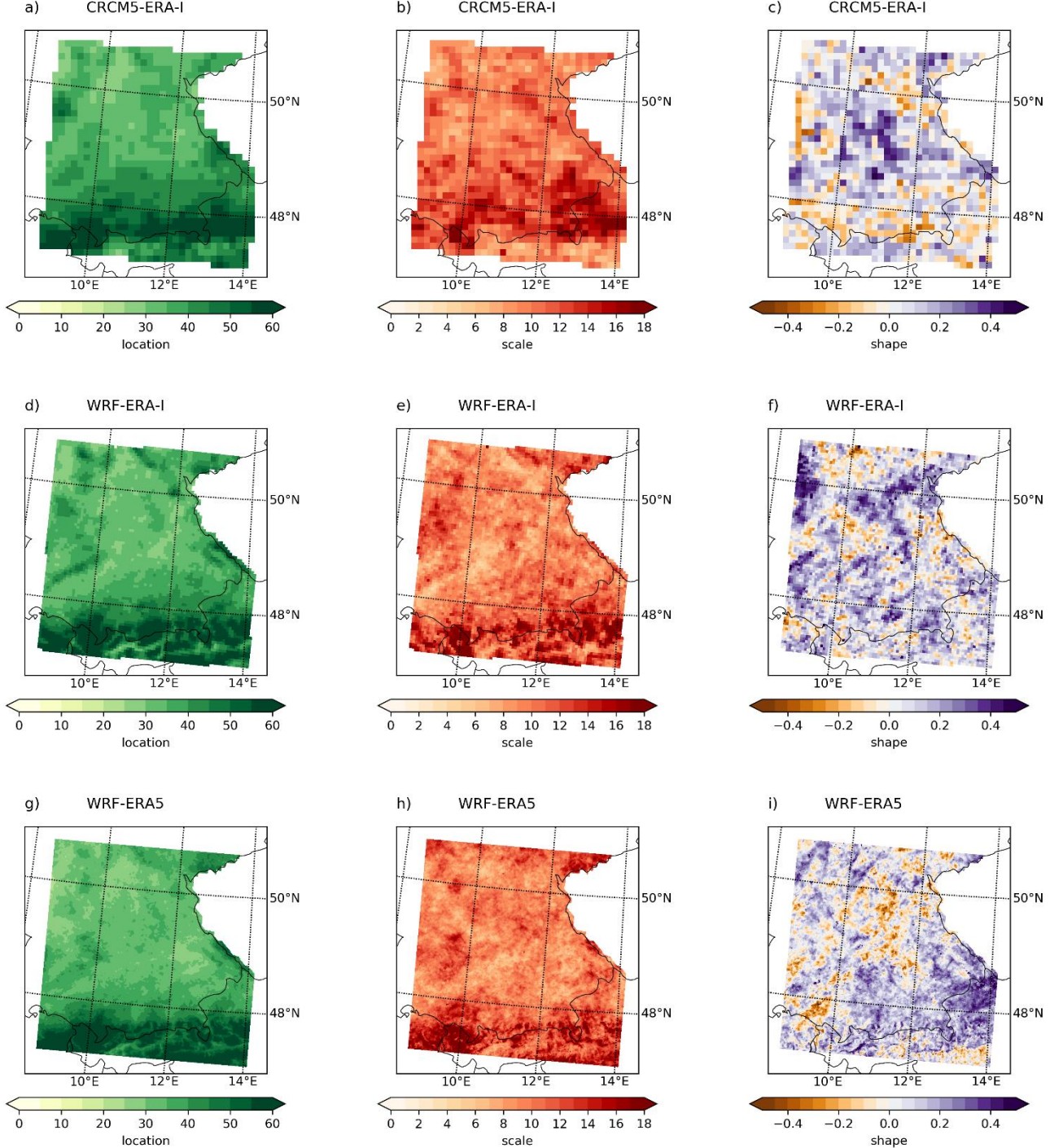


**Figure 3: Location, scale, and shape parameters of the GEV-LMOM approach based on the CRCM5-ERA-Interim (a-c), WRF-ERA-Interim (d-f), and WRF-ERA5 (g-i).**

## 3.2 Peak over threshold

The second classical approach peak over threshold (POT) tries to overcome the drawbacks of the BM sampling as all values $s$ above a threshold $u$ are sampled as extreme values (Balkema and de Haan, 1974; Picklands, 1975). Therefore, multiple values per block are allowed. However, additional restrictions have to be introduced to ensure approximately independent samples. To prevent successive data points from being sampled that originate from one persistent rainfall event, the time series has to be de-clustered. Therefore, a temporal threshold $t_{decluster}$ is chosen and all values within the duration of $t_{decluster}$ around the sampled extreme value are discarded (Coles, 2001).

For the POT approach, the exceedances $y = s - u$ are sampled for the threshold $u$ and samples $s > u$. Thereby, the number of exceedances per year is assumed to follow a Poisson distribution (Davison & Smith, 1990). The exceedances $y$ of the POT threshold $u$ are described by the two-parameter Generalized Pareto (GP) distribution (Davison and Smith, 1990, Martins and Stedinger, 2001). The corresponding CDF is given by

$$H(y;\ \xi) = \begin{cases} 1 - (1 + \frac{\xi y}{\beta})^{-1/\xi}, \xi \neq 0, \beta\ > \ 0, y > 0 \\ 1 - \exp\left(\frac{-y}{\beta}\right), \xi = 0, \beta\ > \ 0, y > 0 \end{cases}, \tag{3}$$

where $y$ defines the precipitation excess over the threshold $u$ of the POT sampling. The scale parameter β and shape parameter ξ describe the spread and tail behaviour of the GP distribution (Coles, 2001).

The GEV and the GP frameworks can be expressed by the other one as the GP distribution corresponds to the tail distribution of the GEV (Coles, 2001; Goda, 2011; Serinaldi and Kilsby, 2014).

For the POT approach, the daily rainfall time series is de-clustered applying a conservative threshold $t_{decluster}$ of 5 days. Typical continental cyclones are found to last up to 2.25 days in Bavaria, whereas van Bebber type Vb cyclones can last up to 3 days (Hofstätter et al., 2018; Mittermeier et al., 2019). Hence, the threshold of 5 days ensures approximately independent samples. Precipitation intensities are assumed to be extreme when exceeding the threshold given by 90 events per 30-year period. This threshold has also been selected by Berg et al. (2019). Statistical properties of the thresholds are given in Table 1 for all three RCM setups. Trends are excluded in the same way as for the GEV-LMOM approach. For sample sizes of n > 50, Delicado and Goria (2008) and Madsen et al. (1997) recommend Maximum Likelihood Estimation (MLE) as optimization algorithm to fit an extreme value distribution. Following this recommendation, MLE is applied to fit the GP distribution to the 90 samples per grid cell using the software package by Gilleland and Katz (2016). The goodness of fit is assessed in the same way as for the GEV-LMOM approach (see Fig. S7) leading to a rejection of 0.15 % of all fits. An exemplary fit for the grid cell of Munich is shown in Figure S4. This approach is referred to as GP-MLE.

**Table 1: Statistical properties of the POT threshold *u* for the three different model setups in the study area.**

| Setup | Minimum | Maximum | Mean | Median |
|-------|---------|---------|------|--------|
| CRCM5-ERA-I | 18.7 mm | 63.8 mm | 29.4 mm | 27.0 mm |
| WRF-ERA-I | 18.1 mm | 71.8 mm | 28.6 mm | 25.8 mm |
| WRF-ERA5 | 12.7 mm | 63.1 mm | 27.1 mm | 23.4 mm |

### 3.3 Metastatistical extreme value framework

For both classical approaches only a limited number of samples contributes to the database of extreme values. A newer

approach by Marani and Ignaccolo (2015) samples all "wet" events assuming that the information of these "ordinary" values can be used to estimate the distribution of extreme values. Thereby, wet events are defined by a threshold $t_{wet}$. It has been successfully applied for extreme daily precipitation by Zorzetto et al. (2016).

The approach by Marani and Ignaccolo (2015) features the Metastatistical Extreme Value (MEV) distribution. They propose a framework supposing that the "meta-statistic" of the rainfall sums of wet events per year contains information about the

intensity of extreme events. They assume the sampled wet days $> t_{wet}$ to be independent following that the probability distribution of maxima $\zeta_m$ can be expressed as $\zeta_m(x) = F(x)^{n_j}$, where $n_j$ is the number of wet events in a year and $F(x)$ is a distribution describing the rainfall sums of these events. Based on the results of Wilson and Toumi (2005), the distribution of rainfall sums during wet days per year is found to follow a distribution with an exponential tail. They expressed precipitation as the product of mass flux, specific humidity and precipitation efficiency. Following statistical relationships, they concluded

that the tail of the distribution of the product of these three random variables is given by a stretched exponential form. Marani and Ignaccolo (2015) and Zorzetto et al. (2016) apply a Weibull distribution to describe this relationship. Hence, Weibull parameters have to be estimated for each year based on all wet events of a year. The MEV-Weibull CDF is given by

$$\zeta_m(x) = \frac{1}{M}\sum_{j=1}^{M}\left\{1 - \exp\left[-\left(\frac{x}{c_j}\right)^{w_j}\right]\right\}^{n_j}, c_j > 0, w_j > 0, \qquad (4)$$

where j is the year (j = 1, 2, …, M), and $n_j$ is the number of wet events in year j. $C_j$ and $w_j$ describe the scale and shape of the Weibull distribution (Marani and Ignaccolo, 2015).

Wet days are defined by exceedance of the threshold $t_{wet} = 1$ mm d$^{-1}$ in accordance with WMO guidelines (Klein-Tank et al., 2009). This also accounts for the behaviour of RCMs to produce too many very low-intensity precipitation days ("drizzle-effect"; Gutowski et al., 2003). As the MEV framework requires the ordinary wet events to be independent (Miniussi et al.,

2020) and temporal autocorrelation of rainfall over mountainous areas tends to be higher (Marra et al. 2021), the autocorrelation of daily rainfall is analysed following Marra et al. (2018; see Fig. S5). In the study area, multi-day precipitation

events are common especially at the mountain slopes (Kunz and Kottmeier, 2006; Pöschmann et al., 2021). Therefore, the temporal autocorrelation is calculated for lag times up to 30 days. The autocorrelation between 10 and 30 days drops to very low values and can be assumed to represent noise without any statistical or meteorological correlation (Marra et al., 2018).

The 75th quantile of this long-lag noise is chosen as "noise threshold". The minimum distance allowed between ordinary events equals the time lag when the autocorrelation first drops below the noise threshold. Hence, the minimum time interval between ordinary wet events may vary within the grid cells, but the independence of the events is ensured by this methodology. The Weibull distribution is fitted to the annual wet events by means of the probability weighted moments method (PWM, Greenwood et al., 1979) following Zorzetto et al. (2016). Here, the MLE is not used as estimation method, as the number of

wet events per year amounts to 40 events on average due to the de-clustering to remove the temporal autocorrelation. For small sample sizes, the MLE estimator for Weibull parameters is known to be biased (Ross, 1996), whereas the PWM delivers unbiased estimations (Heo et al., 2001). The MEV fitting procedure and the calculation of return levels is carried out using the Python software package mevpy (Zorzetto, 2021). The goodness of fit of the annual wet events applying the Weibull distribution is tested with a Kolmogorov-Smirnov test at the significance level of $\alpha = 0.05$, where the *p*-value is adjusted for

multiple testing. Less than 0.06 % of all 30 annual fits per grid cell are rejected for all climate models. This approach is referred to as MEV-PWM. An exemplary comparison of the resulting return level curve to the empirical annual maxima is shown in Figure S6 for the grid cell of Munich.

## 4 Results

### 4.1 Evaluation of 10-year return levels

All approaches and their performance metrics are summarized in Table 2. A mapped comparison of the 10-year return levels calculated via GEV-LMOM based on the three different RCM setups is given in Figure 4. For a better visualization, the observational product is bilinearly interpolated to the respective RCM grid. The following metrics are calculated for the original data (see Fig. S14 for the natively resolved observational product). The observational product shows the highest rainfall intensities above 100 mm d$^{-1}$ at the northern slopes of the Alps. The low mountain ranges of the Bavarian Forest,

Swabian Jura, Odenwald and Ore Mountains also induce enhanced intensities between 70 mm d$^{-1}$ and 100 mm d$^{-1}$. The lowest return levels are observed in the north of the study area amounting to intensities below 50 mm d$^{-1}$ (Fig. 4b, e, h). The 12-km resolution CRCM5-ERA-I can reproduce the general spatial pattern with a Spearman rank correlation of $\rho = 0.72$ (Fig. 4a). The return levels are generally overestimated north of 48° N and underestimated south of 48° N as well as in the Ore Mountains (Fig. 4c). The spatially averaged bias amounts to +6.6 %. The range of simulated rainfall return level intensities is similar to

the observations for the whole study area (Fig. 5a) as well as for the southern alpine part (Fig. 5d). However, the histogram also reveals that the bias stems from simulating too few grid cells with return level intensities between 50 mm d$^{-1}$ to 60 mm d$^{-1}$ and too many grid cells with return levels at intensities of 70 mm d$^{-1}$ to 90 mm d$^{-1}$ (Fig. 5a).

The 10-year return levels based on the WRF-ERA-I at 5 km resolution can recreate the spatial pattern of the observations with a Spearman correlation of $\rho = 0.82$ (Fig. 4d). The higher intensities due to the orographic precipitation at the lower mountain

ranges and their spatial patterns are reproduced, though the intensity around the Bavarian Forest is underestimated. In the alpine area, the WRF-ERA-I simulates higher intensities than observed, especially in the Alps southeast of the Inn valley. However, the results also show a very pronounced orographic signal with low return levels in the major Alpine valleys, which has also been described by Warscher et al. (2019). The overall bias amounts to +4.7 %. The histogram of simulated return levels is similar to the observed histogram (Fig. 5b), however, the very-high intensities above 110 mm d$^{-1}$ in the alpine area

are overrepresented. Also, the range of simulated return levels extends to over 140 mm d$^{-1}$ (Fig. 5e).

**Table 2: Summary of the applied RCM setups and EVT approaches. Performance metrics of the comparison to observational 10-year return levels are given in terms of spatially averaged bias and spatial correlation (Spearman). The entries are sorted first by resolution and then amount of bias.**

| RCM | Reanalysis data | Spatial resolution | Convection | Sampling | EVD | EVD parameter optimization | Bias | Spatial correlation |
|---|---|---|---|---|---|---|---|---|
| WRF | ERA5 | 1.5 km | Explicitely calculated | Block maxima | GEV | L-Moments | +2.4 % | 0.82 |
| WRF | ERA5 | 1.5 km | Explicitely calculated | Block maxima | GEV | Maximum likelihood estimation, fixed shape parameter | +2.5 % | 0.79 |
| WRF | ERA5 | 1.5 km | Explicitely calculated | Peak over threshold | GP | Maximum likelihood estimation | +2.9 % | 0.81 |
| WRF | ERA5 | 1.5 km | Explicitely calculated | All wet events | MEV | Probability weighted moments | -5.8 % | 0.84 |
| WRF | ERA-Interim | 5 km | Parametrized | Block maxima | GEV | Maximum likelihood estimation, fixed shape parameter | +4.1 % | 0.84 |
| WRF | ERA-Interim | 5 km | Parametrized | Block maxima | GEV | L-Moments | +4.7 % | 0.82 |
| WRF | ERA-Interim | 5 km | Parametrized | Peak over threshold | GP | Maximum likelihood estimation | +5.4 % | 0.81 |
| WRF | ERA-Interim | 5 km | Parametrized | All wet events | MEV | Probability weighted moments | -7.1 % | 0.88 |
| CRCM5 | ERA-Interim | 12 km | Parametrized | All wet events | MEV | Probability weighted moments | -2.6 % | 0.84 |

| CRCM5 | ERA-Interim | 12 km | Parametrized | Block maxima | GEV | L-Moments | +6.6 % | 0.72 |
|-------|-------------|-------|--------------|--------------|-----|-----------|--------|------|
| CRCM5 | ERA-Interim | 12 km | Parametrized | Peak over threshold | GP | Maximum likelihood estimation | +7.0 % | 0.72 |
| CRCM5 | ERA-Interim | 12 km | Parametrized | Block maxima | GEV | Maximum likelihood estimation, fixed shape parameter | +7.3 % | 0.74 |

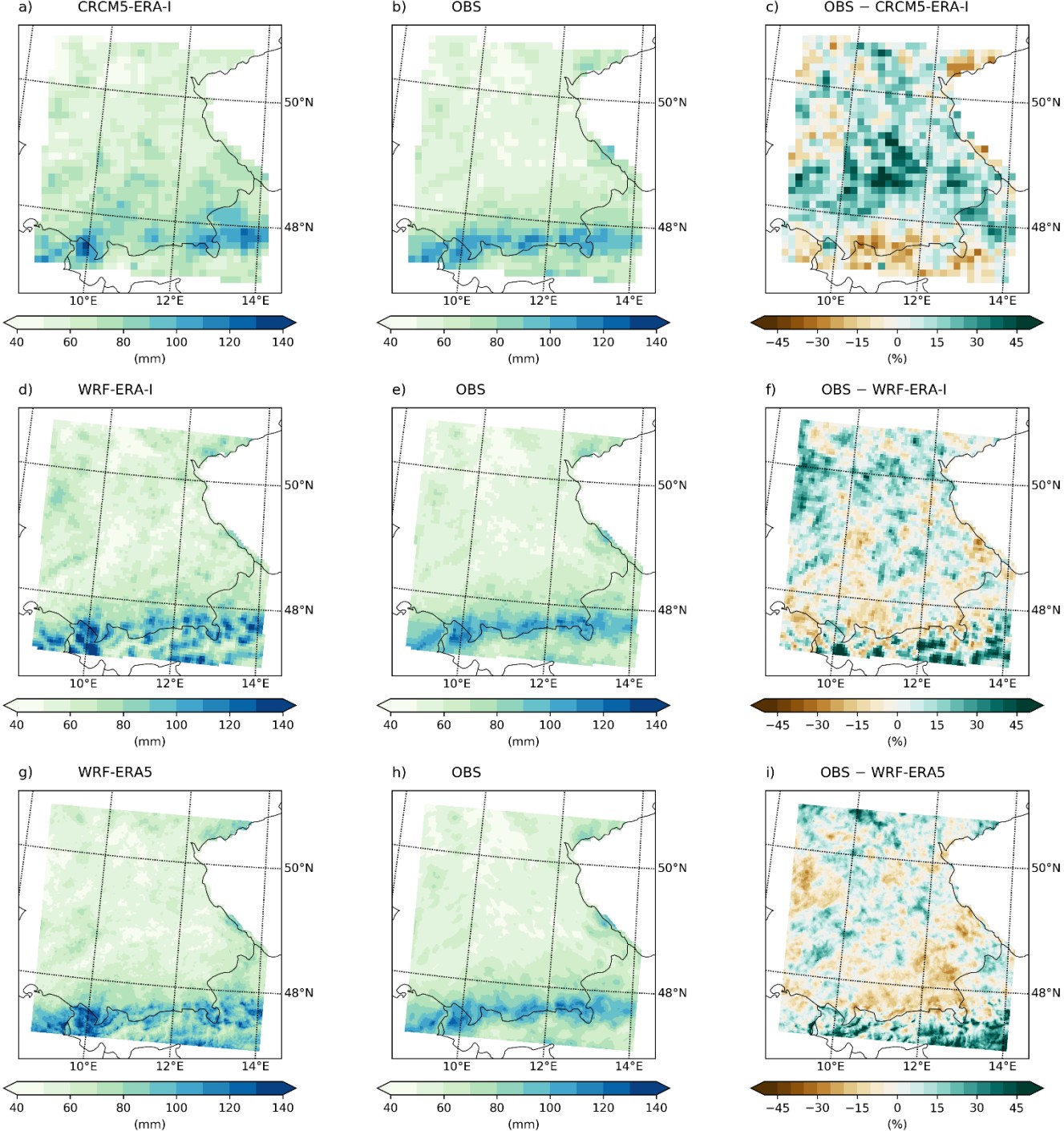


**Figure 4: 10-year rainfall return levels applying GEV-LMOM based on the CRCM5-ERA-Interim (a), WRF-ERA-Interim (d), WRF-ERA5 (g). The middle column (b, e, h) shows the observational product bilinearly interpolated to the respective climate model grid. The right column (c, f, i) provides the percentage difference calculated as climate model return level minus observational return level.**

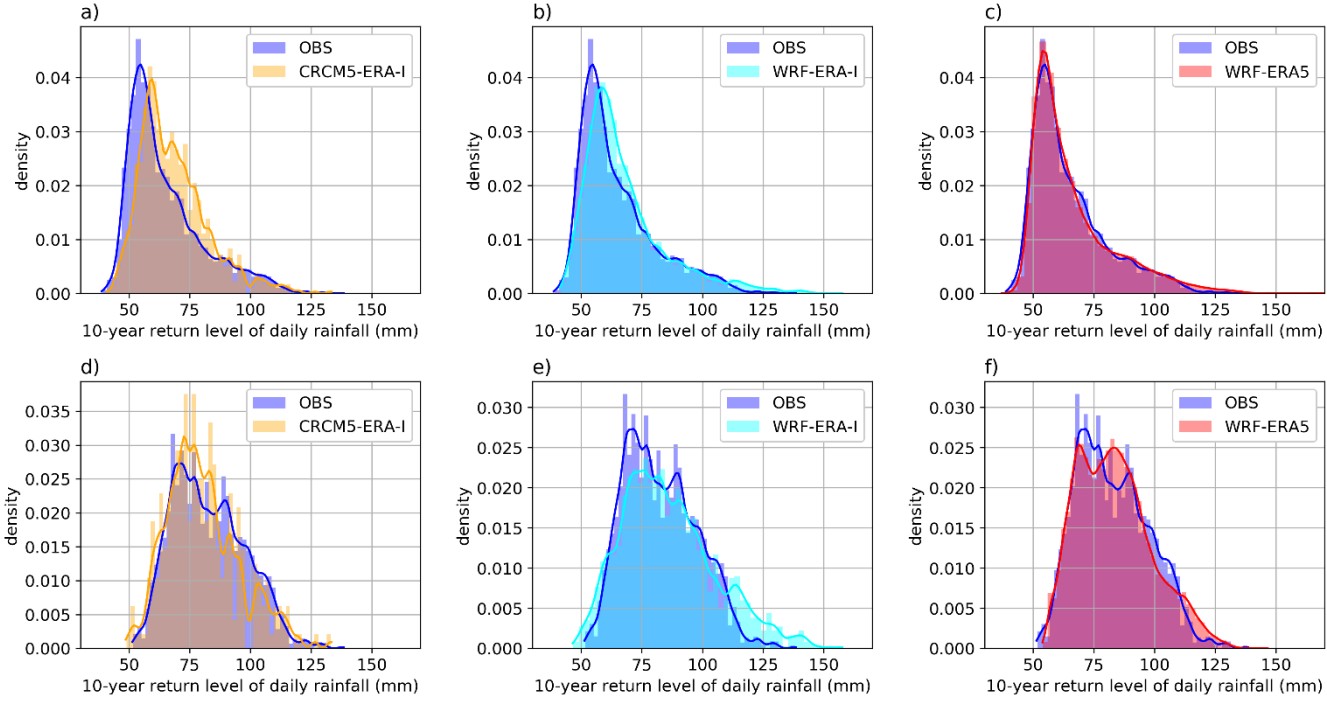

**Figure 5: Histograms of the resulting 10-year return levels in the whole study area (a-c) and the alpine area south of 48° N (d-f). Gaussian kernel density estimates are plotted to enhance the readability.**

The 10-year return levels based on the WRF-ERA5 show a generally similar spatial pattern to the WRF-ERA-I (Figs. 4g and 4d). The spatial pattern of orographic precipitation around the low mountain ranges is recreated, whereby the intensities at the Bavarian Forest and the Odenwald are underestimated. The return levels in south-eastern Bavaria are underestimated as well. As the WRF-ERA-I, also the WRF-ERA5 simulates high return levels above 100 mm d$^{-1}$ in the Alps southeast of the Inn valley. This results in a Spearman correlation of $\rho = 0.82$. The spatial average of the bias amounts to +2.4 %. The range and

distribution of the simulated return levels is very close to the observations for the whole study area (Fig. 5c) as well as south of 48° N (Fig. 5f).

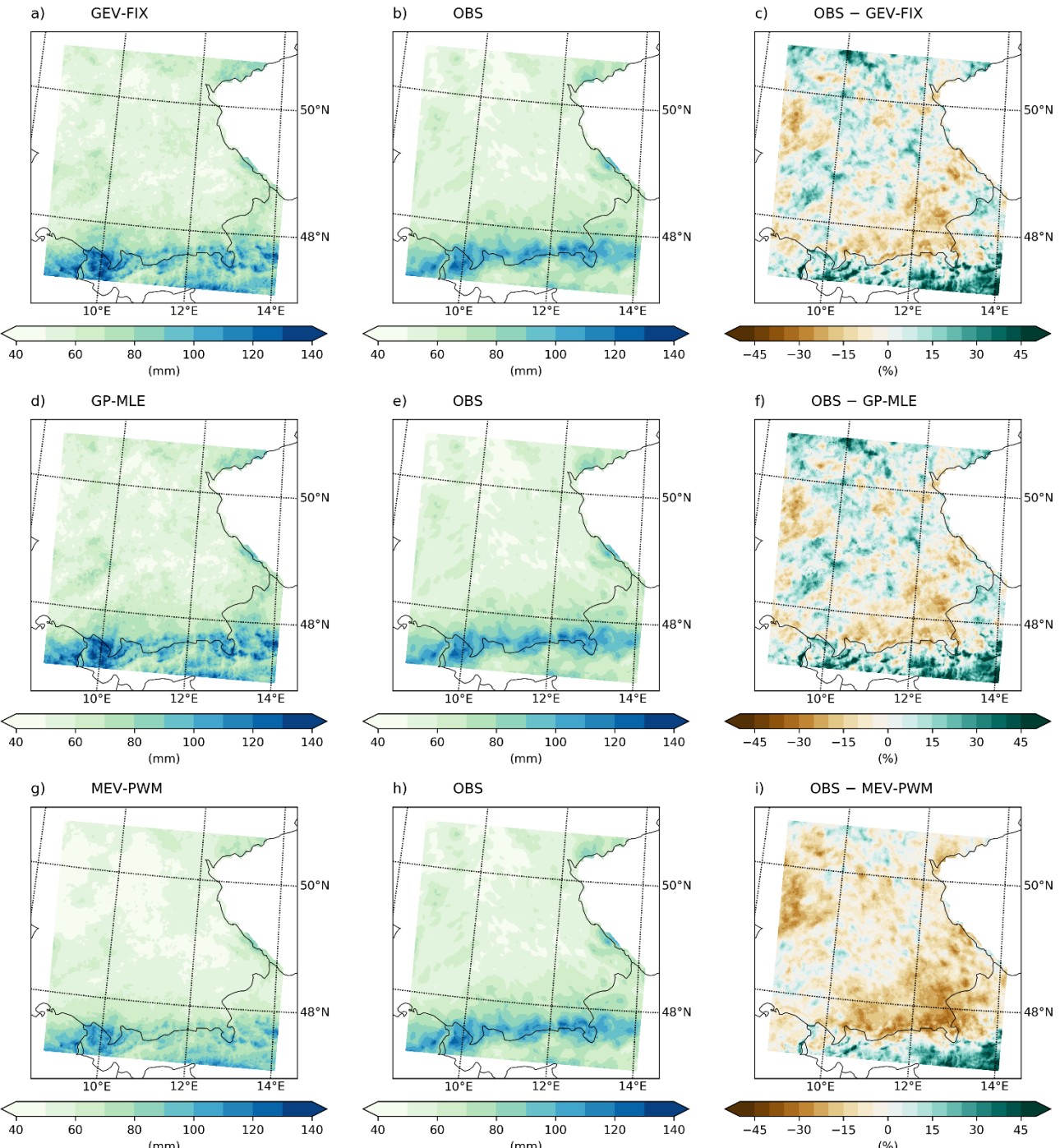

**Figure 6: 10-year rainfall return levels based on the WRF-ERA5 featuring GEV-FIX (a), GP- MLE (d), and featuring MEV-PWM (g). The middle column (b, e, h) shows the observational product interpolated to the WRF-ERA5 grid. The right column (c, f, i) provides the percentage difference calculated as climate model return level minus observational return level.**

Figure 6 compares the three different EVT approaches GEV-FIX, GP-MLE, and MEV-PWM based on the WRF-ERA5. The intensities as well as the resulting spatial distribution of the GEV-FIX and the GP-MLE are very similar to the GEV-LMOM (Fig 4g, 6a and 6d). The spatial correlation between GEV-FIX and the observations amounts to $\rho = 0.79$ and the overall bias to +2.5 %. For the GP-MLE, the spatial correlation is $\rho = 0.81$ and the overall bias is +2.9 %. The MEV-PWM method also shows a similar spatial pattern (Fig. 6g), which is slightly more homogeneous than the GEV-LMOM or GP-MLE. The 10-year return levels based on the MEV-PWM are estimated generally lower than by the classical approaches. The spatial correlation between MEV-PWM and the observations amounts to $\rho = 0.84$ and the overall bias to -7.8 %. The 10-year return levels of the other combinations of climate model and EVT approach that are not presented in the main article are provided in the Supplementary Materials (Figs. S8-S10).

## 4.2 Evaluation of 100-year return levels

The summary of the performance of all RCM setups and EVT approaches at the reproduction of the 100-year return level is given in Table 3. Figure 7 shows the 100-year return levels based on the GEV-LMOM approach compared to the observational product. The observational return levels reach values of 150 mm d$^{-1}$ up to 200 mm d$^{-1}$ at the northern slopes of the Alps. In the Bavarian Forest and Ore Mountains return levels of 100 mm d$^{-1}$ up to 150 mm d$^{-1}$ are observed. The low mountain ranges of the Swabian Jura and Odenwald show intensities between 90 mm d$^{-1}$ and 120 mm d$^{-1}$. The lowest return levels of 50 mm d$^{-1}$ to 60 mm d$^{-1}$ are observed in the plains and leeward sides of the low mountain ranges (Fig. 7b, e, h). The 100-year return levels based on the CRCM5-ERA-I and the GEV-LMOM approach show a similar spatial pattern as the 10-year return levels, however, very high intensities over 180 mm d$^{-1}$ are generated in the centre of the study area around 49° N. These values correspond to the high shape parameter values at these grid cells (see Fig. 3c). Apart from these areas, this approach produces too many grid cells in the range of 90 mm d$^{-1}$ to 140 mm d$^{-1}$ and too few in the range of 60 mm d$^{-1}$ to 90 mm d$^{-1}$ (see Fig. 8a). In the Alps, the simulated 100-year return levels slightly underestimate the observations (Fig. 8d). Overall, this approach cannot well reproduce the general spatial pattern ($\rho = 0.38$). The spatial average of the bias amounts to +15.5 %.

The GEV-LMOM based on the WRF-ERA-I also suffers from single grid cells with unrealistically high return levels (> 200 mm d$^{-1}$ north of the Alps) due to a high shape parameter (Fig. 3f). The return levels in the areas of all low mountain ranges except the Bavarian Forest are overestimated, especially in the Odenwald in the north-west of the study area (see Fig. 7f). This general overestimation is also visualized in the histogram of Figure 8b. In the Alps, the 100-year return levels also show the strong orographic signal of the WRF-ERA-I leading to a greater variance of return levels than observed (Fig. 8e). The spatial pattern is recreated with $\rho = 0.55$ and the overall bias amounts to 17.8 %.

Applying the GEV-LMOM on the WRF-ERA5 also leads to single grid cells with very high return levels scattered over the study area (Fig. 7g), where the shape parameter is greater than 0.5 (Fig. 3i). Apart from these locations, the spatial features of the observed 100-year return level are well reproduced ($\rho = 0.66$). On average, the intensities are overestimated (Fig. 8c)

amounting to a bias of 13.3 %. In the Alpine area, the simulated rainfall return levels show a greater mean and variance (Fig. 8f).

The application of the further three EVT approaches is shown (Fig. 9) and discussed based on the WRF-ERA5. The full overview of all climate models and EVT approaches is provided by Figures 7, S11, S12, and S13. Fixing the shape parameter to $\xi = 0.114$ can eliminate the single grid cells with unrealistic return levels (compare Figs. 7g and 9a). The general spatial pattern is similar, however, the GEV-FIX leads to less variance over the whole study area, as the shape parameter is restricted to one value. Hence, areas with very low intensities based on GEV-LMOM are higher based on GEV-FIX, and high return

levels of GEV-LMOM are reduced by GEV-FIX (Figs. 7g and 9a). The comparison to the observational product (Fig. 9c) results in a spatial correlation of $\rho = 0.70$ and an overall bias of 12.9 %.

The GP-MLE approach also generates single scattered values with higher intensity, e.g., in the Swabian Jura and in the north-west of the study area (Fig. 9d). These intensities are not as high as for the GEV-LMOM, but these cells differentiate inhomogeneously from their respective neighbouring cells. Generally, the spatial patterns and the range of return level values

is similar to the GEV-LMOM. Hence, also the performance metrics in terms of spatial correlation ($\rho = 0.63$) and overall bias (11.9 %) are close to the metrics of GEV-LMOM.

The 100-year return levels based on the MEV-PWM approach differ from the other EVT approaches in terms of the spatial pattern and rainfall intensities. The spatial pattern north of 48°N is very similar to the observations with slight underestimations around the Odenwald and the pre-alpine areas in south-eastern Bavaria. However, in the alpine foreland and norther slopes of

the Alps, the rainfall intensities are underestimated. In sum, this results in a spatial correlation of $\rho = 0.73$ and an averaged bias of 2.7 %.

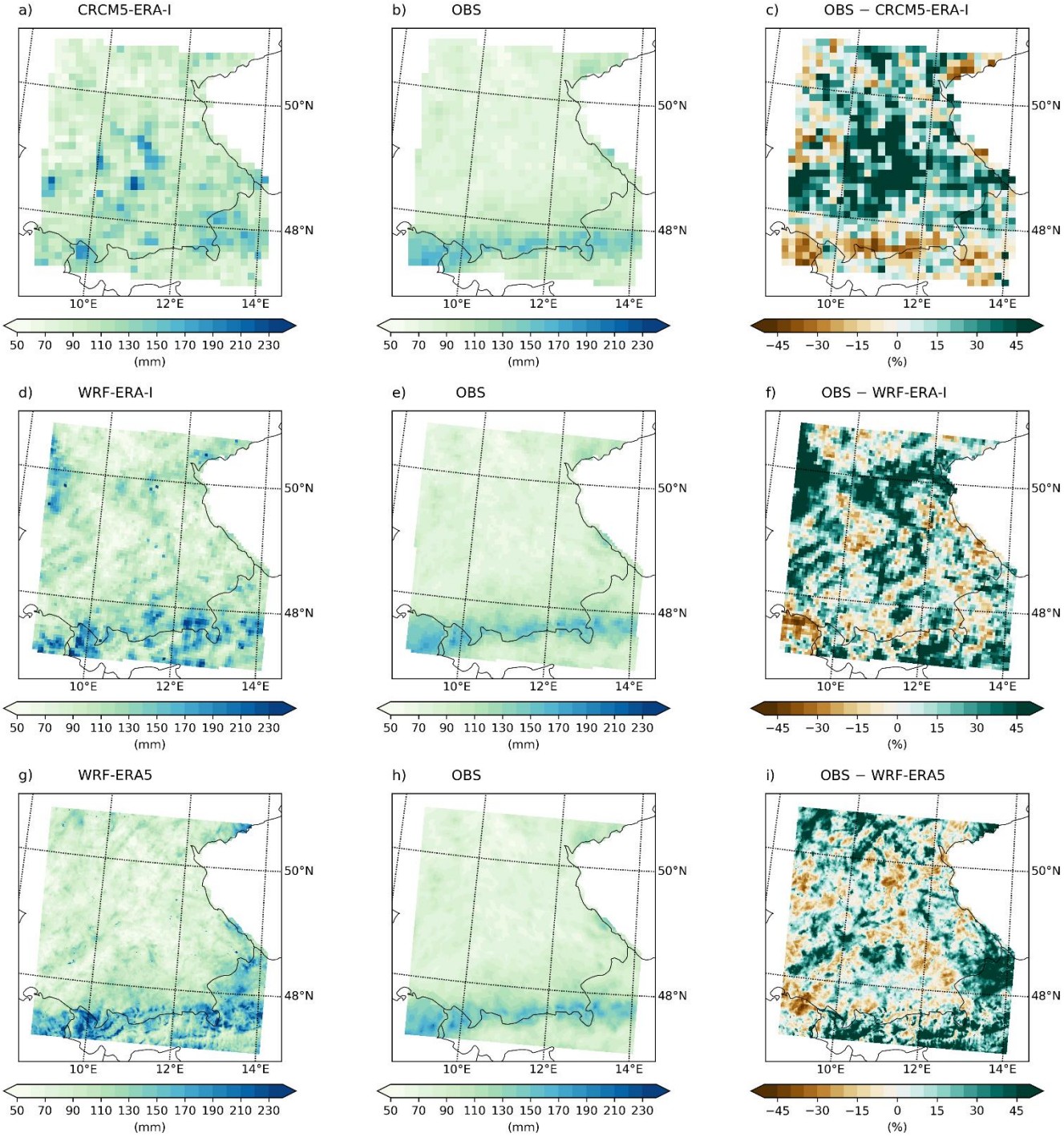


**Figure 7: 100-year rainfall return levels applying GEV-LMOM based on the CRCM5-ERA-Interim (a), WRF-ERA-Interim (d), WRF-ERA5 (g). The middle column (b, e, h) shows the observational product bilinearly interpolated to the respective climate model grid. The right column (c, f, i) provides the percentage difference calculated as climate model return level minus observational return level.**

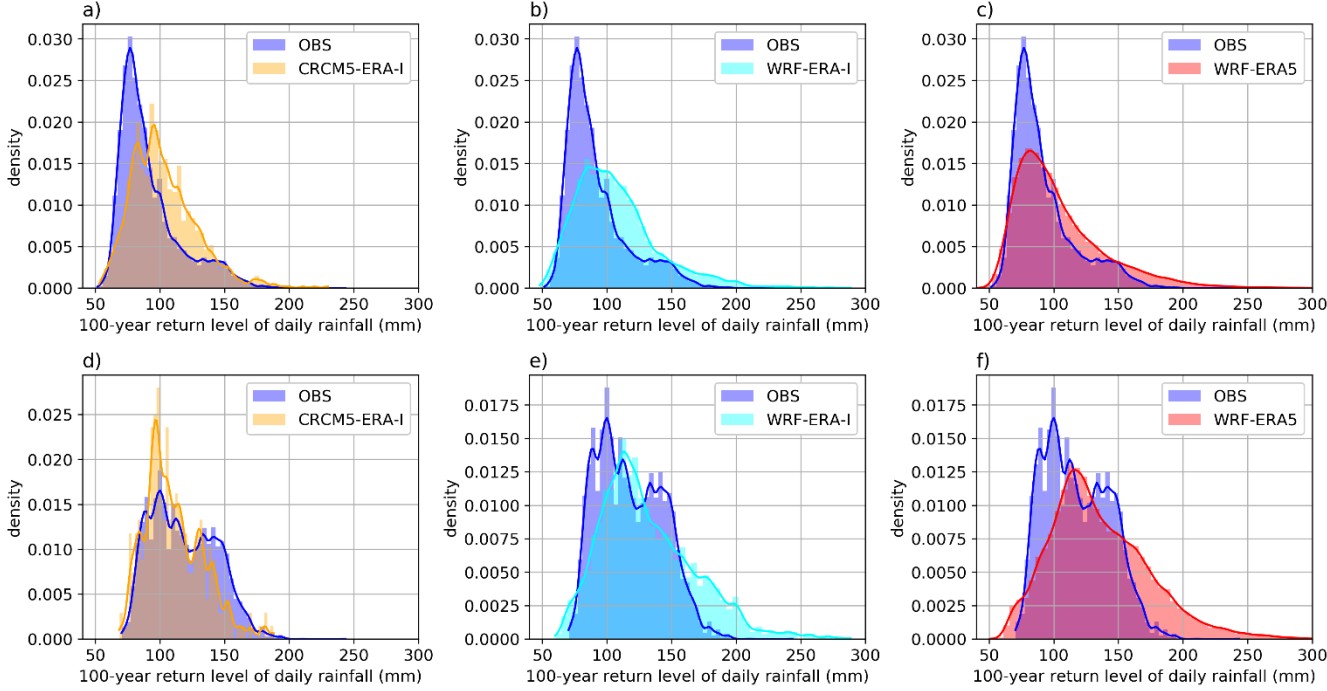

**Figure 8: Histograms of the resulting 100-year return levels in the whole study area (a-c) and the alpine area south of 48° N (d-f). Gaussian kernel density estimates are plotted to enhance the readability.**

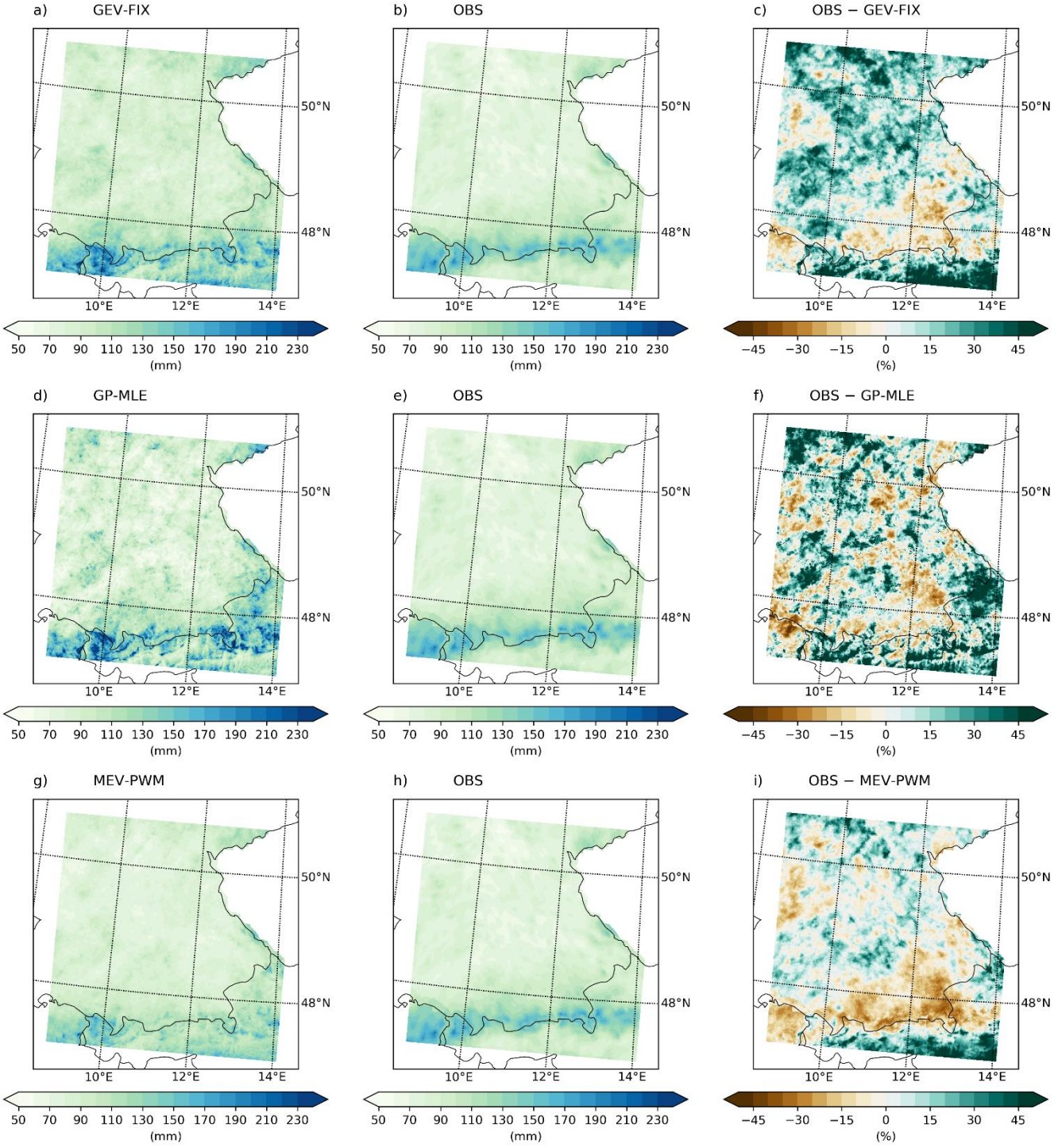


**Figure 9: 100-year rainfall return levels based on the WRF-ERA5 featuring GEV-FIX (a), GP- MLE (d), and featuring MEV-PWM (g). The middle column (b, e, h) shows the observational product interpolated to the WRF-ERA5 grid. The right column (c, f, i) provides the percentage difference calculated as climate model return level minus observational return level.**

**Table 3: Summary of the applied RCM setups and EVT approaches. Performance metrics of the comparison to observational 100-year return levels are given in terms of spatially averaged bias and spatial correlation (Spearman). The entries are sorted first by resolution and then amount of bias.**

| RCM | Reanalysis data | Spatial resolution | Convection | Sampling | EVD | EVD parameter optimization | Bias | Spatial correlation |
|---|---|---|---|---|---|---|---|---|
| WRF | ERA5 | 1.5 km | Explicitely calculated | All wet events | MEV | Probability weighted moments | +2.7 % | 0.73 |
| WRF | ERA5 | 1.5 km | Explicitely calculated | Peak over threshold | GP | Maximum likelihood estimation | +11.9 % | 0.63 |
| WRF | ERA5 | 1.5 km | Explicitely calculated | Block maxima | GEV | Maximum likelihood estimation, fixed shape parameter | +12.9 % | 0.70 |
| WRF | ERA5 | 1.5 km | Explicitely calculated | Block maxima | GEV | L-Moments | +13.3 % | 0.66 |
| WRF | ERA-Interim | 5 km | Parametrized | All wet events | MEV | Probability weighted moments | -1.3 % | 0.72 |
| WRF | ERA-Interim | 5 km | Parametrized | Block maxima | GEV | Maximum likelihood estimation, fixed shape parameter | +13.9 % | 0.76 |
| WRF | ERA-Interim | 5 km | Parametrized | Peak over threshold | GP | Maximum likelihood estimation | +14.7 % | 0.57 |
| WRF | ERA-Interim | 5 km | Parametrized | Block maxima | GEV | L-Moments | +17.8 % | 0.55 |
| CRCM5 | ERA-Interim | 12 km | Parametrized | All wet events | MEV | Probability weighted moments | +4.2 % | 0.72 |

| CRCM5 | ERA-Interim | 12 km | Parametrized | Peak over threshold | GP | Maximum likelihood estimation | +12.6 % | 0.37 |
|---|---|---|---|---|---|---|---|---|
| CRCM5 | ERA-Interim | 12 km | Parametrized | Block maxima | GEV | L-Moments | +15.5 % | 0.38 |
| CRCM5 | ERA-Interim | 12 km | Parametrized | Block maxima | GEV | Maximum likelihood estimation, fixed shape parameter | +17.7 % | 0.62 |

**5 Discussion**

Generally, the high values of the Spearman's rank correlation as well as the visual comparison to the observational product (Fig. 4) prove that all three RCM setups are able to capture the topographic and climatic differences within the study area. Furthermore, the overall low bias of the 10-year return levels indicates that the complex climate of heavy daily precipitation is reproduced by the climate models. For the 100-year return levels, the choice of the EVT approach has a greater impact on the performance metrics.

Both, the overall bias, and the spatial correlation of 10-year and 100-year return levels imply that the WRF setups at 5 km and 1.5 km spatial resolution can slightly better reproduce the observed return levels than the broader-resolution CRCM5-ERA-I (Tables 2 and 3). There, both WRF setups show similar performance metrics. However, this equivalent performance may be caused by the spatial resolution and spatial representativeness of the observational data (see Fig. S14 for the native resolution). The German dataset is natively resolved at roughly 8 km, and the Austrian dataset at 6 km, whereas the Swiss data are given

by single gauges interpolated via ordinary kriging. Hence, small-scale spatial features below such resolution cannot be evaluated by comparison to this observational product. Comparing the WRF-ERA-I and WRF-ERA5 (Figs. 4d, 4g, 7d, and 7g) reveals a similar spatial pattern, where the higher-resolved WRF-ERA5 can especially add more topographically driven spatial variability in the Alps.

**5.1 Uncertainties of the observational datasets**

As the German, Austrian, and Swiss data are based on rain gauge measurements, these data are subject to the usual measurement inaccuracies leading to an underestimation of rainfall (Westra et al., 2014). For flat areas in Germany, this deviation is estimated about 5 % during summer (Richter, 1995). In mountainous areas, this deviation is expected to increase due to higher wind speeds. According to Sevruk (1981) it amounts to 7 % for Switzerland during summer. In addition to these

systematic underestimations, different rain gauge types yield varying rainfall measurements inducing additional uncertainty

(Vuerich et al., 2009). This applies for the different meteorological networks in the study area (Kainz et al., 2007; Frei and Schär, 1998; Rauthe et al., 2013, Zolina et al., 2008).

Apart from these measurement errors, the gridded return level products suffer from a limited number of rain gauges (see Section 2.2), which also differ within their temporal coverage (Isotta et al., 2014). However, not only the number of stations, but also their spatial representativeness is important for an appropriate interpolation from point-wise measurements to gridded estimations (Ahrens, 2006). In mountainous areas, the spatial representativeness of a station is even more limited due to the heterogeneous topography. In addition, the station distribution with elevation is not representative as well. Due to easier maintenance conditions more stations are located in valleys than on the tops of the mountains (Ahrens, 2006; Sevruk, 1997) leading to an underestimation for spatially interpolated rainfall in these areas (Isotta et al., 2014). Although the monitoring network density in the Alps makes this one of the best-monitored regions with complex topography, Isotta et al. (2014) estimate the "real" spatial resolution of the observations to be $10 - 25$ km. The regionalization of these point-wise measurements induces additional uncertainties. For the German dataset, the orography is employed as additional variable to interpolate the return levels (Malitz and Ertel, 2015 following Bartels, 1992). Due to the limited spatial representativeness of the rain gauges in the Alps, the weather model OKM at 1.5 km resolution (Lorenz and Skoda, 2001) was used to support the spatial interpolation of the Austrian return level data (BMLRT, 2018; Kainz et al., 2007). Thereby, not only the spatial distribution of return levels was supported by the weather model simulations, but also the intensity of the resulting design rainfall return levels. The return levels based on observations only are classified as "probably too low" due to the spatial distribution of the rain gauges, whereas the weather model return levels are estimated to be "probably too high" (BMLRT, 2006; 2018). Hence, the resulting design rainfall return level is a weighted averaging combination of the measured rainfall intensities and the intensities simulated by the weather model (BMLRT, 2006). This leads to the conclusion that the deviations of the 10-year and 100-year return levels between the WRF setups (see Figs. 4f, 4i, 7f, and 7i) and the observational data in the Austrian Alps may be caused by the limited spatial representativeness of the measurement stations.

For the Swiss data, ordinary kriging is applied to regionalize the available pointwise return levels. As different interpolation methods yield differing results (Hu et al., 2019), this processing step induces additional uncertainty.

In summary, it can be stated that the study area offers a good temporal and spatial coverage of measurements, especially compared to other regions in Europe (Poschlod et al., 2021), which are, however, subject to the uncertainties mentioned above. Additionally, uncertainties due to the application of different EVT approaches contribute to the overall uncertainty, which are discussed in Section 5.3 as they apply for both observations and climate model data.

Hence, the 10-year return levels (Fig. 4b, e, h) and 100-year return levels (Fig. 7b, e, h) provide the best guess based on observations, but are still matter to substantial uncertainties, especially in the Alps.

**5.2 Uncertainties of the RCM datasets**

Generally, climate model simulations of historical conditions are subject to two major uncertainty factors (Hawkins and Sutton, 2009). Due to the chaotic nature of atmospheric processes the climate system is governed by internal variability. These non-

linear dynamics lead to the behaviour of the system that slightly differing starting conditions may result in significantly

differing climate realizations (Deser et al., 2012). However, in this study the degree of internal variability is constrained as the RCMs are forced by reanalysis data. The large-scale atmospheric flows are imposed by the lateral boundary conditions, and therefore this source of internal variability is not present in these three RCM setups (Christensen et al., 2001). Still, RCMs are governed by smaller-scale atmospheric variability. Alexandru et al. (2007) have shown that a 20-member RCM ensemble of the CRCM driven by the same lateral boundary conditions with slightly perturbed starting conditions leads to a reasonable

spread of simulated precipitation. Even seasonal weather model forecast simulations, which are initialized every month, still show variability, especially for precipitation extremes (Kelder et al., 2020; Thompson et al., 2017). Hence, internal variability cannot be fully excluded as uncertainty source.

Since models can only represent a simplified image of reality, the structure of climate models leads to the second major uncertainty factor. Even though mainly physically-based, RCMs make use of parametrizations with a differing degree of

complexity (Jerez et al., 2013). Model uncertainty includes all limitations of the climate model setup such as model-inherent simplifications, parameterizations and schemes, the lateral boundary conditions, nesting, nudging, spin-up times, and spatial resolution.

Multi-model experiments using the same boundary and starting conditions yield deviating simulations of the climate (Holtanová et al., 2019; Solman et al., 2013). Yet also the same model applying differing physics options and parametrization

schemes can lead to significant variability in the model results (Laux et al., 2019). Hence, climate model setups can be optimized by choosing different model options and schemes and comparing the simulations to observed climate conditions. For the WRF-ERA-I setup, this has been carried out for the whole domain covering central Europe following Wagner et al. (2018; Warscher et al., 2019). The CRCM5-ERA-I and the WRF-ERA5 setups are based on former applications of the respective climate model in different domains. Adapting the applied options to the study area could potentially improve the

model performance (Collier and Mölg, 2020).

Additional uncertainty is induced by the boundary conditions, as different reanalysis datasets show considerable deviations to each other (Keller and Wahl, 2021). Here, two different reanalysis datasets at 75 km (ERA-I) and 30 km spatial resolution (ERA5) covering differing time periods are used to drive the RCMs. Stucki et al. (2020) argue that the difference of the driving conditions regarding the spatial resolution can alter the simulation results, especially over complex terrain.

The different time windows (1980 – 2009 for ERA-I and 1988 – 2017 for ERA5) lead to different events being sampled. Due to the small sample size, this variance can also lead to deviations in the resulting return levels.

The overall differences between the three RCM setups indicating model uncertainty are less apparent in the resulting return levels than in the evaluation of individual extreme events. For the close reproduction of extreme rainfall events, Stucki et al. (2020) have shown that initialization of the RCM briefly before the respective events improves the performance at recreating

rainfall intensities. Here, the RCMs are run in climate mode featuring transient 30-year simulations (CRCM5-ERA-I, WRF-ERA-I) or annual initialization (WRF-ERA5). It cannot be expected that single extreme events are closely reproduced due to the internal variability. Hence, such comparison is not appropriate to evaluate the skill of the model, but to visualize the

differences due to internal variability and model uncertainties. Therefore, the daily rainfall intensities of the two extreme events in May 1999 and August 2005 are given in the Supplementary Materials (Figs. S15 and S16). Furthermore, such a comparison makes it clear that the compared setups are climate model setups and not weather model setups, despite the high spatial resolution (Kelder et al., 2020). While the simulation of individual extremes can differ greatly, the 10-year and 100-year return levels as a climatic indicator for extreme precipitation show a high degree of agreement (see Figs. 4 and 7). This suggests that despite all the simplifications and differences leading to model uncertainty, the models can reproduce the climatic character of extreme precipitation in the study area.

### 5.3 Uncertainties due to EVT

The concept of classical EVT (see Sections 3.1 and 3.2) holds under rather restrictive assumptions (Papalexiou et al., 2013) and each step featuring the choice of the distribution and fitting the distribution parameters induces additional uncertainty (Miniussi and Marani, 2020). For the GEV approach, Eq. (1) holds for a large number of samples $n$ (ideally the sample size $n \rightarrow \infty$). In practice, the limited available time series make it very difficult to determine whether the distribution of extreme samples is close to its asymptotic GEV limit (Cook and Harris, 2004; Koutsoyiannis, 2004; Miniussi and Marani, 2020).

The POT approach partly overcomes the limitation of very low sample sizes by using the threshold $u$ to define extreme events. However, the choice of this threshold is crucial as the assumptions of a Poisson arrival of exceedances $y$ as well as the GP distribution of these exceedances only hold for a threshold $u$ ensuring both the sampled events to be "extreme" and a large number of samples $n$ (Picklands, 1975, Miniussi and Marani, 2020).

Furthermore, uncertainty is induced by the parameter optimization of the respective EVD to adapt the theoretical EVD to the extreme precipitation samples, even though appropriate methods are chosen (see Section 3.3; Muller et al., 2009). Assessing the goodness of fit by quality measures or statistical testing (e.g. the Anderson-Darling test) can lower the uncertainty due to the aforementioned assumptions. However, the goodness of fit can only assess the quality of the fit between the theoretical EVD and the empirical distribution of the samples. It cannot evaluate if the samples are a "good representation" of possible extreme rainfall events within the boundaries of internal variability of the climate system.

Uncertainty is therefore apparent as the different sampling approaches, EVDs, and fitting methods may lead to differing estimations of rainfall return levels (Lazoglou and Anagnostopoulou, 2017). For the GEV-LMOM (Fig. 4g) and GP-MLE (Fig. 6d) based on the WRF-ERA5 the mean absolute deviation (MAD) between the 10-year return levels based on both approaches amounts to a spatial average of 1.7 %. The MAD between the respective 100-year return levels amounts to 8.0 %. Hence, despite different sampling, distributions, and fitting methods, the results are close to each other on average. Larger deviations occur at single grid cells.

However, both classical approaches still suffer from drawbacks. Papalexiou and Koutsoyiannis (2013) as well as Serinaldi and Kilsby (2014) argue that producing stable fits of the shape parameter of the GEV and GP distributions needs larger sample sizes than typically available. They have shown that the estimation of the shape parameter of the GEV and GP distributions is

dependent on the sample size, whereby it is also influenced by the geographical location. Papalexiou and Koutsoyiannis (2013) propose to restrict the shape parameter of the GEV to a window described by a normal distribution around the mean value of 0.114 or to apply a fixed shape parameter of $\xi = 0.114$. Indeed, the shape parameter of GEV-LMOM shows a heterogeneous spatial distribution with small patches of positive and negative values for all three RCM setups (Figs. 3c, f, i). This spatial distribution can be interpreted as "noise" due to too low sample sizes, where the 30 annual maximum precipitation events do not fully represent the range of possible extreme precipitation within the boundaries of internal climate variability. The distribution of the shape parameter $\xi$ based on all three RCM setups is centred around a value close to 0.114 (see Fig. S1c). However, Papalexiou and Koutsoyiannis (2013) suggest that 99 % of the distribution should be between 0 and 0.225, whereas the distribution of all three RCM setups reveals a larger spread. When restricting the shape parameter to $\xi = 0.114$, however, the 10-year return levels only differ very slightly from the GEV-LMOM resulting in an average MAD of 3.4 % (see Fig. 6a). As $\xi$ only defines the tail of the distribution it is more relevant for longer return periods. Hence, high values of the shape parameter (Figs. 3c, f, i) strongly influence the resulting 100-year return level. The outcome of this issue shows up as unrealistically high rainfall intensities (Figs. 7a, d, g) at single grid cells. The GP-MLE approach also suffers from this problem to a lesser extent (Fig. 9d). The fixed shape parameter prevents this issue (Figs. 9a, S11). However, restricting the shape parameter also restricts the flexibility of the GEV, which results in a smaller range of 100-year return levels. The low return levels in the plains and leeward areas are therefore slightly overestimated. However, the resulting 100-year return levels show a higher degree of homogeneity than the 100-year return levels of GEV-LMOM or GP-MLE (compare Fig. 9a to 7g and 9d). In addition to unstable parameters fits, the sampling strategies of both classical EVT approaches only use a small fraction of available data. In this study, only 0.3 % (GEV) and 0.8 % (GP) of the daily precipitation sums from the RCMs are sampled. Especially with respect to short available observation time series, but also with respect to the extensive computational power and related costs of such high-resolution climate models, this sampling is a waste of valuable information (Miniussi and Marani, 2020). The sampling of the MEV approach overcomes this limitation and uses the information of rainfall intensities of all wet days as well as the frequency of these days. This is found to result in more stable fits (Zorzetto et al., 2016). Zorzetto et al. (2016) concluded that the MEV outperforms the classical GEV approach due to the more stable parameter fits if the return period exceeds the length of the available samples. Furthermore, they found that the MEV is better than the GEV at predicting return levels if the EVT models are calibrated on samples, which are independent from the samples used to calculate the return levels. In this study, the MEV-PWM return levels are on average lower than the return levels based on the other EVT approaches (Tables 2 and 3). While this leads to an average underestimation of the observational product at the 10-year return levels, the MEV can outperform the other EVT approaches at the 100-year return levels. In terms of the spatial correlation, the MEV-PWM leads to superior results overall than the other approaches for both calculated return levels. The moderate to strong underestimation of rainfall intensities at the 100-year return period in the area of the Pre-Alps and northern Alps (Fig. 9i) is mainly attributed to the climate model data, as all EVT approaches yield lower intensities in this area as well (Figs. 7i, 9c, 9f). The other EVT approaches "compensate" these low intensities by their tendency to overestimate the 100-year return levels (see Figs. 7i and 9). Schellander et al. (2019) apply the MEV optimized via PWM and GEV optimized via

MLE on 55 rain gauges with more than 100 years of measurements in Austria, which is partly covered by the study area. They split the data, where up to 50 years are used to calibrate the EVT models. The remaining data are the basis to calculate the return levels, which are used for the evaluation of the GEV and MEV. They find that the MEV can outperform the GEV for return periods of 30 years or longer, when less than 30 years of data are available. For the two cases of this study (sample size of 30 years and return periods of 10 and 100 years), they report a slightly superior performance of the GEV for 10-year return levels and a slightly superior performance of the MEV for 100-year return levels. In sum, the results of their study are in line with the findings of this study, even if the differences between GEV and MEV in this investigation are a little more pronounced, especially for the 100-year return period.

## 6 Conclusion

Various combinations of high-resolution regional climate models driven by reanalysis data and state-of-the-art EVT approaches have been explored to reproduce 10-year and 100-year return levels of daily rainfall. The 5 km WRF-ERA-I setup reveals added value in terms of spatial correlation and bias compared to the lower-resolution 12 km CRCM5-ERA-I. The very high resolution 1.5 km WRF-ERA5, accompanied by an explicit simulation of convective processes, can only slightly improve the performance metrics. This is possibly since the observational product is resolved natively between 6 km and 8 km. Hence finer-scale spatial features cannot be evaluated by such comparison. Despite the improvement of overall performance metrics, local biases in the order of 30 to 40 % still remain. Therefore, the criticism of the practitioners that was expressed for the CRCM5 return levels from Poschlod et al. (2021; see Sect. 1) would also be present for the return levels shown here.

The resulting 10-year return levels based on the four different applied EVT approaches show good agreement to each other and to the observational product. This suggests that the methodological uncertainty for return levels of moderate extremes is relatively low. However, if return periods outside the sample size are to be extrapolated, the estimation uncertainty of the shape parameter governing the tail of the GEV and GP distributions becomes more important. The 100-year return levels based on the GEV-LMOM and GP-MLE suffer from single grid cells with unrealistically high return levels due to high estimations of the shape parameter. Two approaches are studied to overcome this uncertainty. The GEV with fixed shape parameter shows 100-year return levels, whose performance metrics are almost equivalent to the three-parameter GEV optimized via L-moments. However, the resulting return levels are homogeneous and do not show any unrealistic outliers. The rather new EVT approach by Marani and Ignaccolo (2015) featuring the MEV distribution leads to a slight underestimation of 10-year return levels but produces the best results for the 100-year return period. This methodology shows great potential for extrapolation of longer return periods due to the larger sampling and, therefore, increased stability of the fits (Schellander et al., 2019; Zorzetto et al., 2016).

The question remains to be answered as to what the findings of this study can contribute to in practice.

First, in regions with a low density of rain gauges, such RCM setups can contribute to a homogeneous spatial estimation of return levels. Even in regions, where the rain gauges cannot represent the spatial heterogeneity, RCMs can be applied to support

observational products. This is already being done in Austria using a convective-permitting weather model, and the results of this study reinforce such use of regional climate models. It is also conceivable to use the high-resolution spatial patterns of CPMs as an auxiliary variable for the interpolation of the return levels based on measured data (e.g. via kriging with external drift; Haberlandt, 2007; or spatial GEV models; Davison et al., 2012; or a spatial representation of the simplified MEV; Schellander et al., 2019). A visualization of a simple combination approach for such a subsequent enhancement of 100-year return levels is provided in Figure 10. Therefore, the differences at each grid cell between the observational product and the WRF-ERA5 MEV-PWM are smoothed with a Gaussian filter and again added to the climate model return levels. However, this rather naive approach only serves to provide a visual impression of a possible enhancement.



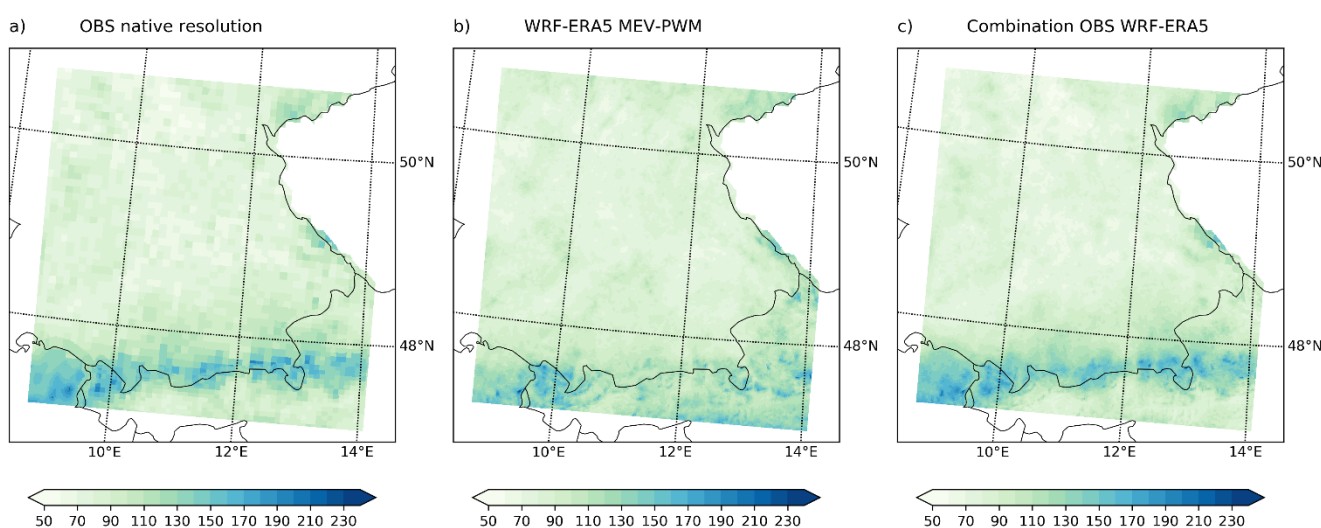

Figure 10: (a) Observational 100-year return levels based on the German (8 km), Austrian (6 km), and Swiss (interpolated via ordinary kriging) data at original resolution. (b) 100-year return levels based on the WRF-ERA5 applying the MEV-PWM approach. (c) Combination of (a) and (b) by applying a Gaussian filter on the differences.

Second, different EVT approaches are explored based on 30 years of data with daily rainfall. For moderate extremes (10-year return level), the differences between the EVT approaches are minor. Due to the slight underestimation of the MEV-PWM, GEV and GP approaches can be recommended for such applications. For return periods, which are longer than the available data, the estimation uncertainty of the shape parameter of the GEV and GP distributions induces unrealistic return level values at single grid cells. Fixing the shape parameter can prevent this issue. However, the MEV framework using the information of all ordinary wet events produces stable fits and shows the best performance at the reproduction of 100-year return levels. It is recommended for applications, where the return period needs to be extrapolated.



Further conclusions regarding the future use of RCMs follow from these findings. Third, large ensembles of RCMs can be set up to increase the sample size within the boundaries of the internal climate variability. On the one hand, increased sample sizes

lower the uncertainty related to EVT, on the other hand large ensembles enable to quantify uncertainties due to internal

variability (Poschlod et al., 2021).

Fourth, RCMs driven by global climate models following different emission scenarios allow to simulate climate change induced alterations of return levels (Ban et al., 2020; Poschlod and Ludwig, 2021). Even though an increase in extreme precipitation intensities is known for decades (Trenberth et al., 2003), there is a lack of operational implementation and adaptation. In 2004, a climate change surcharge of a flat +15 % on top of the 100-year flood return level was introduced in

Bavaria for the planning of flood protection facilities (LfU, 2021). Indeed, trends in the magnitude of floods in Bavaria can be detected (Blöschl et al., 2019). However, such an adaptation for extreme rainfall is missing so far, even though there is much higher consensus in the scientific community about the increase in extreme rainfall intensities than about the increase in floods (Sharma et al., 2018, Merz et al., 2021).

Despite all model-specific uncertainties, the evaluation of RCMs in this study proved that they are suitable to reproduce daily

extreme precipitation intensities over complex terrain.

**Data availability**

The observational rainfall return level data are available at the German weather service (DWD, 2020), the Federal Ministry of Agriculture, Regions and Tourism Austria (BMLRT, 2020), and MeteoSwiss (MeteoSwiss, 2021).

The daily precipitation of the WRF-ERA-I and WRF-ERA5 are publicly available at Warscher (2019) and Collier (2020), which is cordially acknowledged. The CRCM5-ERA-I data are available on https://climex-data.srv.lrz.de/Public/ERA_driven_run/pr/.

**Competing interests**

The author declares that he has no conflict of interest.

**Financial support**

The author acknowledges the support within the project StarTrEx (Starkniederschlag und Trockenheitsextreme; Heavy precipitation and drought extremes; Az. 81-0270-82467/2019) by the Bavarian Environment Agency.

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
