# Peer review of "Using high-resolution regional climate models to estimate return levels of daily extreme precipitation over Bavaria"

_Natural Hazards and Earth System Sciences, 2021_

## Author Response (AR1)

Dear Editor and dear Reviewers,

Thank you for your time and efforts to review this manuscript. Your suggestions have improved the study structurally, methodologically and in the findings, for which I cordially thank you! The revision has been carried out generally following the answers to the reviewers' comments (https://doi.org/10.5194/nhess-2021-66-AC1 (**AC1**) and https://doi.org/10.5194/nhess-2021-66-AC2 (**AC2**)). I will briefly list the major changes as overview and then address all comments point-by-point. Please also find the tracked-changes version of the manuscript. However, it is not as easy to read, due to the number of changes and the rearrangements of two sections.

I hope that the revised version of the manuscript and my answers can satisfyingly address your comments and suggestions.

Kind regards,

Benjamin Poschlod

**Major changes:**

- Reordering of the paragraphs in Section 2
- Rearranging Section 3
- Calculation of 100-year return levels and processing the 100-year observational product
- Calculation of all possible combinations of the three RCM setups and four EVT approaches for 10-year and 100-year return levels
- MEV: Analysis of the autocorrelation and de-clustering the ordinary wet events to ensure (approximately) independent events (Fig. S5)
- MEV: Adding a goodness-of-fit test to test the annual fits of the Weibull distribution to all wet events per year
- Exemplary fits at the grid cell of Munich: All approaches and RCM setups are shown including 95%-CI (Figs. S2, S3, S4, S6)
- Goodness of fit of all approaches and RCM setups is presented in terms of the p-values (Fig. S7)

**Point-by-point answers RC1:**

(comments in blue, answers in black; line numbers correspond to the new manuscript version without tracked changes)

Major comments:

1) In the conclusions the author clearly stated the uncertainties arising from different model setups regarding internal climate variability, parametrizations, and further assumptions. Saying so, why did you then choose different RCMs and not only a single one with similar setups, e.g., a COSMO-CLM version in the given (slightly different) resolutions? Furthermore, why did you use ERA-Interim and ERA5 as forcing data and not only the higher resolved and newer ERA5 data for all simulations?

The CRCM5 simulation was chosen as the "reference" RCM simulation due to the previous studies based on this model (Poschlod et al. 2021 & Poschlod and Ludwig, 2021; refs. in the manuscript), which were presented to the Bavarian Environmental Agency.

Therefore, I chose higher resolution simulations from freely available data sources covering the study area with a time period of 30 years driven by reanalysis data. The 5km WRF as representative for high-resolution simulations with parametrization of convection, and the 1.5 km WRF as the highest-resolution simulation known to me without any parametrization of deep and shallow convection.

This is described in **L148-154** in the new manuscript version: "The selection of these three different setups was based on the following considerations: The CRCM5 driven by a global climate model has proven to reproduce rainfall return levels over Europe with good skill (Poschlod et al., 2021). As described in Section 1, the resulting return levels of this RCM driven by a global climate model were presented to local authorities, but local biases prevented further implementation of the results. Therefore, this setup serves as a benchmark. The WRF ERA-INTERIM at 5 km resolution represents a setup optimised for the study area with higher spatial resolution but parameterisation of convection. The WRF ERA5 is the highest resolution setup available with 1.5 km resolution and calculates convection explicitly. All three climate model rainfall data sets are openly available."

2) The author put lots of effort into the homogenization of pointwise observational data sets. There are several high-res gridded precipitation data sets on the market like REGNIE/HYRAS for Germany (1km, Rauthe et al., 2013), RADOLAN (DWD, 1km), or SPARTACUS (Austria, 1km, Hiebl and Frei, 2017). I agree that even at this high resolution these data sets have limitations when it comes to convection. Nevertheless, DWD and ZAMG put a lot of effort into calibrating these data sets not only with ground measurements but also with radar data and vise versa in the case of RADOLAN. So, I assume these data sets have a higher quality than the homogenized point observations by the author and they have a higher resolution which made the validation of the 1.5km WRF model more robust.

In addition to the explanation given in **AC1**, I have made sure with representatives of the Bavarian Environmental Agency (personal communication) that they use KOSTRA for their applications as legal guideline, which is why I kept this observational product as validation measure.

3) When it comes to different extreme value techniques, a proper validation would use every method with every data set and not only a couple of possible combinations like currently presented.

All possible combinations are now calculated. All performance metrics are presented in Tables 1 and 2. The mapped return values of the newly calculated combinations are shown in the Supplement (Figs. S8-13).

4) The authors conclude that RCMs are better in terms of spatial representativeness of return levels. Saying so I expect cross-validation with existing products like KOSTRA for Germany to clearly point out the benefit of RCMs compared to raw or existing gridded observations.

As described in **AC1**, I did not want to state that RCMs are generally better than KOSTRA in terms of spatial representativeness of return levels. In areas with low rain gauge density (e.g. large parts of Scandinavia or eastern Europe), RCMs can support the observations. Or in areas, where rain gauges have low spatial representativity (e.g. in the Alps due to the heterogeneous terrain), RCMs can be useful, which is why Austria supports the observations with the OKM model. To evaluate if RCMs are appropriate for such applications, they are compared to the observations in the study area, where the rain gauge density is very high.

5) The author concentrated on the return level of 10 years and stated that this is the most important value for the targeted applications. At least for the insurance industry, minimum the 100-year return level better the 200-year values (PML200) are the relevant levels. As all results are specifically related to the 10-year level, I am wondering if the methodology can be adapted/used for higher return levels or if further validation/calibration is necessary in that case. I miss some statements on that in the discussion and conclusions sections.

The 100-year return level is calculated for all approaches and compared to the observational product. Generally, the EVT approaches now have a bigger impact on the results than for the 10-year return levels. The RCMs can still reproduce the observations with moderate to good performance at bias and spatial correlation (Table 2). The MEV framework can outperform the other EVT methods for the 100-year return level.
The results are presented in Section 4.2 (**L390-450**) and discussed (**L583-589** and **L600-614**).

Additionally to the major comments above, I have some minor comments [page-line/paragraph]:

[Sect. 1] I recommend clearly state the key research questions you are focusing on in this study. For me, it is not clear what the main aims are.

**L106-109:** "The study tries to answer two main research questions: (1) Can existing RCM setups at higher spatial resolution reduce local biases and improve spatial correlation between the climate model products and the observational product? (2) How large are the differences due to the application of different state-of-the-art extreme value statistical approaches, and which approach is recommended?"

[P3 L81ff] Schröter et al. (2015) analyzed three major flood events in Germany during the past 70 years (1954,2002,2013), which also partly affected your investigation area, concluding that it is not daily/multi-day precipitation amount that triggers major flood events.

**L100f:** "However, the antecedent wetness state of the catchment also plays a major role in the transition of heavy precipitation to floods (Schröter et al., 2015)."

[P3 L88ff] "RCM can bridge the gaps" – what about stochastic weather generator or other approaches? Ehmele and Kunz (2019), for example, introduced a semi-physical, 2D, and high-resolved precipitation model mainly based on orographic precipitation which in a statistical sense, gives good results in terms of return levels even for higher return periods.

**L58ff:** "Ehmele and Kunz (2019) apply a semi-physical two-dimensional stochastic precipitation model to calculate spatial homogeneous return levels over Baden-Württemberg (Germany). However, the model needs to be calibrated with observational data and therefore relies on the high rain gauge density in the area."

[Sect. 2] I recommend a reordering of the paragraphs in this section. As your investigation area is restricted to the given data sets, I suggest first describe the data sets and the investigation area afterward.

The Section has been reordered according to your suggestion.

[Fig.1] Is the study area equal to the model domain? If so, how do you deal with boundary effects?

**L180ff:** "The 1.5 km domain covers 351 × 351 grid cells, whereby the outer 40 cells are discarded on all sides to exclude boundary effects (Collier and Mölg, 2020)."

[P4 L97f] In Fig.2 you give the reference for the data set, I suggest giving it in the text, too.

**L188f:** "The patterns of annual mean precipitation are governed by the complex topography (see Fig. 2; Haylock et al., 2008)."

[Fig.2] Do you have an explanation for the strong "drying" signal in the main Alpine valleys? Please use discrete color separations. See also https://www.nature.com/articles/s41467-020-19160-7

As in **AC1**, I and Warscher et al. 2019 (ref. in the manuscript) have no specific explanation for this behaviour. I mentioned this sensibility to orography and the corresponding "drying" in the valleys in **L351f.**, as it is also pronounced for extreme precipitation: "However, the results also show a very pronounced orographic signal with low return levels in the major Alpine valleys, which has also been described by Warscher et al. (2019)."

Figure 2 (**L200**) is re-drawn using E-OBS and an appropriate color scheme ("viridis" as suggested by https://www.nature.com/articles/s41467-020-19160-7 ).

[Sect. 2.2] So I understand that you estimate daily precipitation or at least 24h sums in the moving window by hourly station data, right? If so, please clarify in the text.

The text is adapted to clarify the adjustment. The DWD uses hourly and daily rain gauges, takes the daily measurements (7:30 AM to 7:30 AM 0:00 AM to 0:00 AM according to the station type), but increases the daily return level values by 14% to generate estimations for 24-h moving windows: **L122f.:** "The resulting daily return levels are increased by 14 % to provide 24-hourly moving window estimates (Malitz and Ertel, 2015)."

Hence, I reverse this step by reducing their 24-hourly levels by 14% again, **L126ff:** "As the daily return levels were beforehand transferred to 24-hourly moving window estimates, I reduce these values by 14 % to obtain daily estimates. This relation between daily fixed windows and 24-hourly moving windows has also been applied by Poschlod et al. (2021) following Barbero et al. (2019) and Boughton and Jakob (2008)."

The Austrian dataset also provides 24-hourly estimates. **L136ff:** "Again, this data product provides moving window 24-hourly estimates, which is why the 24-hourly return levels are adjusted to daily values applying a reduction of 14 % (see Sect. 2.1.1)."

I hope this description can clarify your question.

[P7 L133] "24h RLs are adjusted to daily values using a reduction". I do not understand what this reduction is about. Please clarify this in the text.

See comment above.

[Sect. 2.3] Why did you choose exactly these models and not others? There is a huge variety of RCM in 0.11° resolution within the CORDEX project and also high-resolution simulations mainly Germany and Alpine region in the CORDEX FPS convection project. Furthermore, you used WRF v3.6.1 for the 5km and v4.1 for the 1.5km simulations. Are there major differences between the versions? For consistency, the same model version would be better.

See major comment 1).

[P8 L161ff] For WRF 1.5km, you have 30 simulations with a 1-year length each. Does this have an impact on the comparability with the continuous simulations at coarser resolution?

**L174ff:** "As the model is forced by the lateral boundary conditions at 3-hourly resolution, slicing the simulation period is not assumed to have a systematic impact on the magnitude of rainfall return levels."

[Sect.3] I suggest a reordering here, too. Instead of first describing strategies and distributions and then how they are applied in this study, I recommend a structure like 3.1 BM; 3.2 POT, 3.3 MEV each with a short introduction to the method and then directly saying how you will apply it in this study.

Section 3 is reordered according to your suggestion.

[P9 L180ff] It would be helpful for the reader if you can give typical values or magnitude orders of t_wet and t_decluster.

Now at **L280** and **L311**.

[P9 L192] G is also a CDF, right? Please indicate it.

Indicated at **L212**.

[P12 L242] Can you explain why the low-res simulations have higher return values than the high-res?

As stated in **AC1**, higher resolution does not necessarily lead to higher return levels.

[P13 L277] You mean Fig.5d instead of 5b?

Corrected

[P15 L289] The 5km WRF seems to have a much stronger orographic signal than the 1.5km, especially the "drying" in the main valleys. Is there any explanation for that?

See also comment [Fig. 2]. As in **AC1**, I and Warscher et al. 2019 (ref. in the manuscript) have no specific explanation for this behaviour. I mentioned this sensibility to orography and the corresponding "drying" in the valleys in **L351f.**, as it is also pronounced for extreme precipitation: "However, the results also show a very pronounced orographic signal with low return levels in the major Alpine valleys, which has also been described by Warscher et al. (2019)."

[P15 293] Fig.5b and later 5e?

Corrected

[P15 L301] Sure you mean Fig 3d here?

Corrected to 4d

[P15 L305] Fig.5c and 5f, I guess

Corrected

[P19 L394-398] Maybe I miss something, but I do not get the message from these two paragraphs

I added a recent reference for the uncertainty of the reanalysis data, which shows the differences of varying reanalysis products over Europe (Keller and Wahl, 2021; ref. in manuscript). **L528ff:** "Additional uncertainty is induced by the boundary conditions, as different reanalysis datasets show considerable deviations to each other (Keller and Wahl, 2021). Here, two different reanalysis datasets at 75 km (ERA-I) and 30 km spatial resolution (ERA5) covering differing time periods are used to drive the RCMs. Stucki et al. (2020) argue that the difference of the driving conditions regarding the spatial resolution can alter the simulation results, especially over complex terrain. The different time windows (1980 – 2009 for ERA-I and 1988 – 2017 for ERA5) lead to different events being sampled. Due to the small sample size, this variance can also lead to deviations in the resulting return levels."

These two paragraphs just discuss the reanalysis data as source of uncertainty, as well as the differing time periods.

[Fig. S5+S6] There is data missing for Switzerland and Austria. Why? I thought you have the data for that regions and time periods.

As stated in **AC1**, Austria and Switzerland do not make these daily gridded rainfall datasets openly available. I have return level data for these two countries, but not the daily data. I added this explanation to the Figures (now S15 and S16).

**Point-by-point answers RC2:**

(comments in blue, answers in black; line numbers correspond to the new manuscript version without tracked changes)

Major comments:

1) The use of the high-resolution products (REGNIE, RADOLAN, SPARTACUS) would avoid to homogenize the gauge precipitation values and would make possible a more accurate validation of the RCMs with the finest resolution. Why not considering them?

In addition to the explanation given in **AC2**, I have made sure with representatives of the Bavarian Environmental Agency (personal communication) that they use KOSTRA for their applications as legal guideline, which is why I kept this observational product as validation measure.

2) Why only return level of 10 years? I understand the concern of the author that 30 years of data are few for estimating higher quantiles, but return periods higher than 10 (e.g., 100) years are more relevant for engineering applications/(re)insurance purposes and the challenge is indeed to estimate them with the availability of short time series. How would the estimation of higher return levels compare e.g. with the official ones from KOSTRA? As the manuscript is presented now, the conclusion stated in the abstract "it follows that high-resolution regional climate models are suitable for generating spatially homogenous rainfall return level products" is not fully supported by the analysis, since only the 10-years return levels have been evaluated.

The 100-year return level is calculated for all approaches and compared to the observational product. Generally, the EVT approaches now have a bigger impact on the results than for the 10-year return levels. The RCMs can still reproduce the observations with moderate to good performance at bias and spatial correlation (Table 2). The MEV framework can outperform the other EVT methods for the 100-year return level.
The results are presented in Section 4.2 (**L390-450**) and discussed (**L583-589** and **L600-614**).

3) The study area is characterized by some high-elevated regions affected by orographic precipitation. I'm wondering if using all the values as "ordinary events" in the MEV might not respect the independence hypothesis required by the MEV framework. See for example Marra et al. (2018) and Miniussi et al. (2020) for some discussion on temporal correlation.

I carried out the analysis and de-clustering following Marra et al. (2018). The enhanced methodology is described in **L313-321**: "As the MEV framework requires the ordinary wet events to be independent (Miniussi et al., 2020) and temporal autocorrelation of rainfall over mountainous areas tends to be higher (Marra et al. 2021), the autocorrelation of daily rainfall is analysed following Marra et al. (2018; see Fig. S5). In the study area, multi-day precipitation events are common especially at the mountain slopes (Kunz and Kottmeier, 2006; Pöschmann et al., 2021). Therefore, the temporal autocorrelation is calculated for lag times up to 30 days. The autocorrelation between 10 and 30 days drops to very low values and can be assumed to represent noise without any statistical or meteorological correlation (Marra et al., 2018). The 75th quantile of this long-lag noise is chosen as "noise threshold". The minimum distance allowed between ordinary events equals the time lag when the

autocorrelation first drops below the noise threshold. Hence, the minimum time interval between ordinary wet events may vary within the grid cells, but the independence of the events is ensured by this methodology."

Generally, the MEV results of the 10-year return levels have only changed slightly due to this de-clustering. The homogeneous spatial distribution is still present.

4) Why using a GEV distribution with a constant shape parameter and not, for example, a Gumbel? Previous studies (e.g., Grieser et al. (2007)) have shown that the Gumbel distribution is a good model for precipitation in the Bavarian area, and its location parameter has a strong correlation with altitude, while its scale parameter has a noisy pattern (except for the Bavarian Alps). Moreover, you say that the shape parameter based on all the three RCM setups is centered around a value close to 0.114, in line with the one recommended by Papalexiou and Koutsoyiannis (2013): is this really a fair comparison, as these shape parameter values are already affected by estimation uncertainty?

I answered this question in **AC2** in detail.

Minor comments.

Section 3.

L225: Another title for section 3.3 would be more appropriate

Adapted due to reordering the section.

L226-227: please add a couple of words about the adjustment, so that the reader understands it directly from here without the need to go looking at the reference.

I explained the problem and adjustment in **L221ff:** "The significance level describes the probability rejecting the null hypothesis $H_0$, given that $H_0$ is true. As the statistical test is carried out at $n$ grid cells, $H_0$ would be erroneously rejected at $n \cdot \alpha$ grid cells on average by design of the test setup (Ventura et al., 2004). The rate of these errors is referred to as false discovery rate (FDR; Benjamini and Hochberg, 1995). To control the FDR, the critical $p$-value is adjusted for multiple testing using the approach from Benjamini and Hochberg (1995) following Wilks (2016). $H_0$ is rejected at each grid cell $g$ if the $p$-value of the test $p_g \leq p_{FDR}$, where

$$p_{FDR} = \max_{g=1,\ldots,n} \left\{ g : p_{(g)} \leq \alpha_{FDR} \cdot \left( \frac{g}{n} \right) \right\} \qquad (2)$$

$p_{(g)}$ with $g = 1,\ldots,n$ are the sorted $p$-values of the statistical test for all grid cells $g$ of the study area. For $\alpha_{FDR}$ the value of $2 \cdot \alpha$ is recommended (Wilks, 2016)."

L239: you state that "the location and scale parameter are governed by the topography". From Figure 3 one can notice that the spatial pattern of the location parameter is somehow coherent with topography, but the noise for the scale parameter does not make its pattern straightforward to understand. Maybe also the colors scale is not helping.

I adapted the color scale for the scale parameter and enhanced the description in **L238ff:** "There, the location parameter is governed by the topography (see Figs. 1 and 3a, d, g), where the spatial distribution of these parameters is similar for all three RCM setups. The spatial distribution of the scale parameter also corresponds to the topography but shows more noise.

The spatial distribution of the WRF-ERA-I and WRF-ERA5 are similar and show the highest values of the scale parameter at the northern slopes of the Alps. The orography of the low mountain ranges of the Swabian Jura, Odenwald, Ore Mountains and Bavarian Forest also impacts the spatial pattern of the scale parameter (Figs. 3e and 3h). Lower values are found at the leesides of the low mountain ranges and the inner-alpine dry valleys. The spatial distribution of the scale parameter based on the CRCM5-ERA-I follows the topography less closely and shows an even noisier pattern (Fig. 3b). Some topographical features can nevertheless be recognised, such as the Odenwald and higher values in the Pre-Alps and northern slopes of the Alps."

L240: why a chaotic pattern for the shape parameter? Is it related to the uncertainty that one can get due to the limited series available to estimate it?

I have now brought the explanation forward and added it already in the results section (**L247f**): "This chaotic pattern corresponds to the high estimation variance of the shape parameter based on the limited available sample size of 30 annual maxima."

L259: you mention you made a "goodness of fit" (despite its limitation in prediction) for the GEV and the GP distributions. Have you made a similar analysis also for the Weibull distribution?

It is now added and explained in **L327**: "The goodness of fit of the annual wet events applying the Weibull distribution is tested with a Kolmogorov-Smirnov test at the significance level of $\alpha = 0.05$, where the p-value is adjusted for multiple testing. Less than 0.06 % of all 30 annual fits per grid cell are rejected for all climate models."

The p-values of all GOF tests and all RCM setups are visualized in Fig. S7. However (as you state), these tests are limited in prediction.

L264: in L253-255 you mention that for sample sizes > 50 estimation via ML is recommended. Why then using PWM for the Weibull distribution in the MEV framework?

L321ff: "The Weibull distribution is fitted to the annual wet events by means of the probability weighted moments method (PWM, Greenwood et al., 1979) following Zorzetto et al. (2016). Here, the MLE is not used as estimation method, as the number of wet events per year amounts to 40 events on average due to the de-clustering to remove the temporal autocorrelation. For small sample sizes, the MLE estimator for Weibull parameters is known to be biased (Ross, 1996), whereas the PWM delivers unbiased estimations (Heo et al., 2001)."

Section 4.

L287 and 310 (captions of Figures 4 and 6): "difference calculated as climate model return level minus observational return level" -> difference between the return level from the climate model and the observational one. Why using of the absolute error instead of the relative error?

The relative error is shown and the caption of Figs. 4, 6, 7, 9, and S8-13 is adapted: "The right column (c, f, i) provides the percentage difference calculated as climate model return level minus observational return level."

L454-457: in Zorzetto et al. (2016) the analysis has been made by means of a cross-validation approach, so that the sample used for parameter calibration is independent from the one used for testing the performance of GEV and MEV distributions. When GEV is fitted and tested on the same sample (unless the sample is shorter, i.e. 10-20 years, when issues in the parameter estimation –especially for the shape parameter- might arise), it usually outperforms MEV, but it is not flexible in prediction.

This statement is changed (L597ff): "Furthermore, they found that the MEV is better than the GEV at predicting return levels if the EVT models are calibrated on samples, which are independent from the samples used to calculate the return levels."

I also add the reference to Schellander et al., 2019 (L606ff): "Schellander et al. (2019) apply the MEV optimized via PWM and GEV optimized via MLE on 55 rain gauges with more than 100 years of measurements in Austria, which is partly covered by the study area. They split the data, where up to 50 years are used to calibrate the EVT models. The remaining data are the basis to calculate the return levels, which are used for the evaluation of the GEV and MEV. They find that the MEV can outperform the GEV for return periods of 30 years or longer, when less than 30 years of data are available. For the two cases of this study (sample size of 30 years and return periods of 10 and 100 years), they report a slightly superior performance of the GEV for 10-year return levels and a slightly superior performance of the MEV for 100-year return levels. In sum, the results of their study are in line with the findings of this study, even if the differences between GEV and MEV in this investigation are a little more pronounced, especially for the 100-year return period."

Supplementary material.

FigS2: how are the 95% confidence intervals computed?

You have the example for the Munich grid cell, and only for GEV-LMOM and GP models, why not for GEV-ML and MEV? Moreover, a comprehensive validation of all the extreme value models for the whole area would add value to the analysis.

The Figures have been updated (S2, S3, S4, S6) and the calculation method of the confidence intervals are added in the figure caption. Additionally, the goodness-of-fit test p-values are provided (Fig. S7).
As now all combinations of RCM setups and EVT approaches are calculated, the results (Tables 1 & 2) give a better overview of the performances of the different EVT approaches.

FigS5-S6: now the REGNIE product is shown; why not showing the observation-based product used in the analysis? It would be also useful to evaluate differences among the products (even if for some events only).

As stated in **AC2**, KOSTRA only provides the return level data, but not the daily data.

---

## Author Response (AR2)

Dear Editor and dear Reviewers,

Thank you for your time and efforts to review the revised version of the manuscript.
I will briefly address the remaining comments point-by-point. Please also find the tracked-changes version of the manuscript.

Kind regards,

Benjamin Poschlod

**Point-by-point answers to anonymous referee 1:**

(comments in blue, answers in black)

Line 140: So what is finally meant by daily values? Is it calender day, 00-00UTC/Local Time or anything else?

This sentence is added: "The daily measurement window spans from 05:50 to 05:50 UTC." The Austrian and Swiss daily measurement windows are now defined in the article as well. They differ by 10 minutes each, so there is no big discrepancy for rainfall events at the country borders.

Lines 170ff: You state that uncertainties or biases prevent the data of a previous study forced with GCMs to be included in further decision making etc at local authorities. Is this the main reason why you now change to reanalysis forcing?

It is an "indirect" reason. The study (Poschlod et al., 2021; ref. in article) using the CRCM5 forced by a 50-member single model initial-condition large ensemble has shown that internal climate variability has major impacts on the estimations of return levels. For this study, where I wanted to test higher spatial resolution setups, no large ensembles are available. Hence, driving the RCM by reanalysis data minimizes internal variability as reason for biases.

This is added to the article:

"The CRCM5 driven by a global climate model ensemble has proven to reproduce rainfall return levels over Europe with good skill (Poschlod et al., 2021). However, the study has shown that internal climate variability has major impacts on the estimation of return levels. Using reanalysis data as boundary conditions strongly reduces this source of uncertainty when comparing with observation-based return levels. As described in Section 1, the resulting return levels of this RCM driven by a global climate model ensemble were presented to local authorities, but local biases prevented further implementation of the results. Therefore, the CRCM5 setup serves as a benchmark."

Sect. 2.2.1/2.2.2: WRF-ERA-I is nested from 75x75km² to 45x45km² to 15x15km² down to th 5x5km² resolution. I wonder why you then perform a direct nest from 75x75km² to 0.11° in case of the CRCM-ERA-I run and not again some kind of nesting.

The "nesting strategy" is chosen by the executing modeller group / institute based on their experience. The CRCM-ERA-I run has been set up by my Canadian colleagues Leduc et al. (2019; ref. in article). This step from 75x75km² ERA-I to 0.11° without nesting is common also within EURO-CORDEX (see Kotlarski et al. 2014, table 1: https://gmd.copernicus.org/articles/7/1297/2014/gmd-7-1297-2014.pdf ).

Hence, I add for clarification in the article: "No nesting was applied, as with the RCM setups presented in Kotlarski et al. (2014), which are also driven by ERA-Interim and have a spatial resolution of 0.11°."

Line 280ff and related figures: In some captions you state the CI95 is calculated with 1000 times bootstrapping and in case of Fig.S3 it says "via delta method". Why did you use a different method in this particular case and what can you say about the "accuracy" of both methods?

For this particular case (GEV with fixed shape parameter), the bootstrapping method is not implemented in any R or Python package known to me. Hence, in order to provide confidence intervals, the delta method is applied. The disadvantage of the delta method is the symmetry of the CIs, which is an unrealistic assumption, especially for long return periods. The bootstrapping method shows a slight tendency for the bootstrap sample to generate shorter tails than the true sample distribution (Coles and Simiu, 2003) resulting in slightly more narrow CIs for long return periods (longer than 100 years; Caires, 2007:

https://repository.tudelft.nl/islandora/object/uuid:8d38ef9c-ead4-4b9d-850c-d4dd2e71a34f/datastream/OBJ/download ). So in the case of this study (return periods up to 100 years) I would prefer the bootstrapping over the delta method.

Coles, S., and E. Simiu, 2003: Estimating uncertainty in the extreme value analysis of data generated by a hurricane simulation model. J. Engrg. Mech., 129 (11), 1288-1294.

Fig. 3: Is there any serious explanation why the WRF-ERA5 simulates opposite sign in shape parameters e.g. over the Franconia region?

Generally, the chaotic pattern of the shape parameter for all three setups is governed by estimation uncertainty due to the small sample size. If you compare all three maps, there are areas for all three setups where one setup differs from the other two. The pattern you describe for WRF-ERA5 in Franconia is indeed the most prominent. There is no "physical explanation" (RCM, reanalysis data set) for this behaviour. As "high estimation variance of the shape parameter based on the limited available sample size" is already mentioned in the article, no further comment is added to the article.

Line 335ff: are the given thresholds from the observations or the simulations?

Thanks a lot for pointing at this. These thresholds were given based on the simulation of WRF-ERA5. In the old version of the paper the POT approach was only carried out for this model setup. I add the statistical properties for the threshold values for all three RCM setups by including a small table.

Figure S4: What do you mean with empirical estimated return periods and how are they calculated?

The "empirical return periods" are based on the empirical annual maxima (for GEV and MEV) and the 90 events over the respective threshold (GPD), which are plotted via plotting position formula. With $n$ as sample size, $k$ as order rank and $P_k$ as empirical distribution function (EDF), the EDF can be expressed as $P_k = k/n$ (de Haan & Ferreira, 2006). However, inspired by your

comment, I now follow Makkonen (2006), who strongly recommends the Weibull plotting position formula, where $P_k = k/(n+1)$. The figures in the supplement are adapted (very slightly) and an according comment is added to the captions.

De Haan, L. and Ferreira, A.: Extreme Value Theory: An Introduction, Springer, 436 pp., New York, 2006.

Makkonen, L.: Plotting positions in extreme value analysis, J. Appl. Meteorol. Clim., 45, 334–340, https://doi.org/10.1175/JAM2349.1, 2006.

Table 1+2: I suggest some kind of sorting/ordering in both tables to make it more readable, e.g. by resolution, bias, etc. If you decide to do so, it has to be stated in the caption as well.

The Tables were sorted by the EVT approaches (GEV-LMOM, GP-MLE, GEV-MLE, MEV-PWM). I admit that this order is not intuitive, and therefore I change the order as you suggest (sorted by resolution and then bias).

Thank you for the useful comments and hints. I hope that your questions and suggestions are answered and implemented sufficiently.

Kind regards,
Benjamin Poschlod